# Following the Human Thread in Social Navigation

**Luca Scofano**[1,*]   **Alessio Sampieri**[1,4,*]   **Tommaso Campari**[2,*]   **Valentino Sacco**[1,*]

**Indro Spinelli**[1]   **Lamberto Ballan**[3]   **Fabio Galasso**[1]

[1] Sapienza University of Rome   [2] Fondazione Bruno Kessler   [3] University of Padova   [4] ItalAI

## Abstract

The success of collaboration between humans and robots in shared environments relies on the robot's real-time adaptation to human motion. Specifically, in Social Navigation, the agent should be close enough to assist but ready to back up to let the human move freely, avoiding collisions. Human trajectories emerge as crucial cues in Social Navigation, but they are partially observable from the robot's egocentric view and computationally complex to process.

We present the first Social Dynamics Adaptation model (SDA) based on the robot's state-action history to infer the social dynamics. We propose a two-stage Reinforcement Learning framework: the first learns to encode the human trajectories into social dynamics and learns a motion policy conditioned on this encoded information, the current status, and the previous action. Here, the trajectories are fully visible, i.e., assumed as privileged information. In the second stage, the trained policy operates without direct access to trajectories. Instead, the model infers the social dynamics solely from the history of previous actions and statuses in real-time. Tested on the novel Habitat 3.0 platform, SDA sets a novel state-of-the-art (SotA) performance in finding and following humans.

The code can be found at https://github.com/L-Scofano/SDA.

## 1 Introduction

Traditional navigation techniques within Embodied Artificial Intelligence (EAI) have marked a crucial advancement by introducing robots into real-life environments. However, these techniques have primarily focused on agents traversing vacant spaces. Conversely, the significance of social navigation within EAI has steadily increased. Social navigation entails agents' capacity to navigate human-centric environments while considering human movements and behaviours. These agents need to be able to locate, track, and interact with humans in a safe and socially acceptable manner. Previous studies predominantly characterized Social Navigation as a variation of PointGoal Navigation, wherein agents strive to reach specified destinations while considering human movements (Wijmans et al., 2019b; Ye et al., 2021; Partsey et al., 2022). Habitat 3.0 (Puig et al., 2024), a significant breakthrough in EAI, introduces a lifelike environment seamlessly incorporating human avatars. This integration enables investigating human-agent interactions within a controlled, risk-free, dynamic environment. What sets Habitat 3.0 (Puig et al., 2024) apart is its ability to replicate complex scenarios where human intentions are constantly changing. Nevertheless, this dynamism also presents particular challenges, such as collision avoidance and achieving success in locating and following humans. Finding a person and following them is relevant to human-robot collaboration. For instance, in search-and-rescue operations, a robot may need to find and reliably follow a human responder through unpredictable and hazardous terrain to assist, carry equipment, or relay critical information. Similarly, in assistive robotics, such as elder care, robots must follow caregivers or patients across different rooms in a home, adapting seamlessly to changes in speed, direction, and environment. The complexity of the task stems from difficult long-term human motion prediction and the possible changes of their intentions, from having to navigate unknown and dynamic environments, and from strict safety requirements.

---

*Equal contribution. Corresponding authors, email: luca.scofano@uniroma1.it. Part of this work has been accomplished while Alessio Sampieri was visiting the Panasonic R&D AI Labs (PRDCA), CA, USA

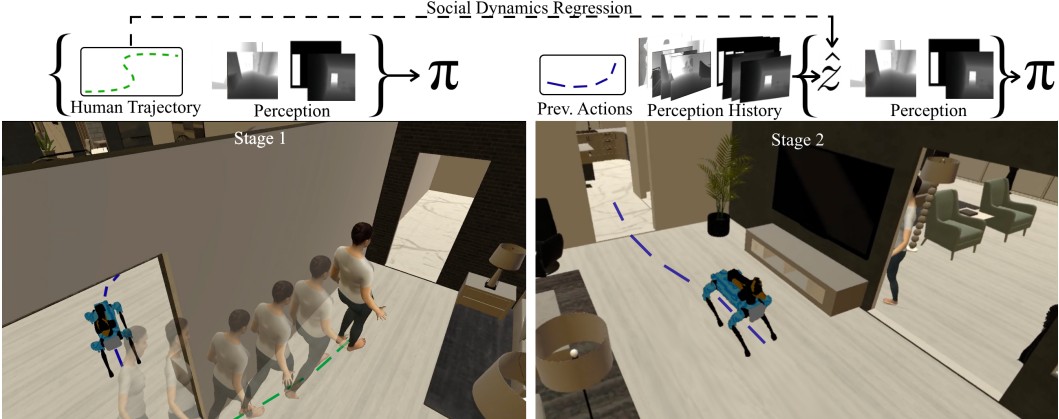

Figure 1: We present our novel Social Dynamic Adaptation model (SDA). The framework involves two stages of training that allow the model to infer, given its past observations and actions, another agent's Social Dynamics. In the first training stage, the model embeds the followed agent's trajectory, which, together with sensor perceptions, compose the input to the model's Social navigation Policy ($\pi$). The knowledge obtained from the human trajectory strongly helps the navigation policy in finding and following an agent. However, this information is often not available during deployment. In the second stage, SDA learns to adapt past statuses and actions, which are always available, to the first stage's Social Dynamics embedding $\hat{z}$. As depicted in the figure, the status contains depth maps and BB detection of the person, if observable from the egocentric robot view. $\hat{z}$ is then paired with current observations as input to the frozen $\pi$.

Despite notable efforts in collision avoidance and safety, most existing methods for Social Navigation either rely on privileged information that is not available in real-world scenarios or do not adequately capture the social dynamics and norms of human behaviour. For instance, (Wijmans et al., 2019b) and (Ye et al., 2021) use a GPS and compass sensor to provide the agent with perfect localization, which might be unrealistic even if using SLAM (Mur-Artal et al., 2015; Engel et al., 2014) methods, whenever the human is not in the line of sight. While (Partsey et al., 2022) and (Yokoyama et al., 2022) do not account for the social factors that influence human behaviour. Therefore, this limitation hinders their practical applicability and adaptability to dynamic environments. (Cancelli et al., 2023), instead, models some social factors in the form of Proximity Tasks but fails to account for the cooperative nature that a social agent must possess, restricting itself to merely avoiding collisions with them. The current SotA model proposed in (Puig et al., 2024) necessitates privileged information, such as humanoid GPS, which offers polar coordinates, detailing the accurate distance and angle of the human from the robot, to attain high-performance outcomes. However, this requirement is highly impractical in real-world environments during inference.

This paper proposes a novel Social Dynamics Adaptation model (SDA), shown in Fig. 2, that effectively solves the robot's awareness of complex human behaviors, even temporarily losing sight of the person and fast robot motion. Specifically, the first stage trains a base policy considering human trajectories encoded into a latent vector. The latent vector is a low-dimensional, nonlinear projection of the human trajectories, and it is trained end-to-end with the base policy to extract the social factors that led the robots to choose better actions. The subsequent supervised adaptation stage regresses this latent vector using only the robot's state and action history. Unlike previous methods, such as (Wijmans et al., 2019b) and (Yokoyama et al., 2022), which often depend on simulated privileged information or simplified social behavior models, our approach adapts to dynamics resembling real-world conditions in real-time. In summary, SDA adapts and accounts for unpredictable human behaviors by exploiting privileged information during training and recovering this fundamental signal during deployment when it is often impractical to compute. Finally, the deployed robot features the motion policy, learned in the first stage, and the social dynamics, inferred from prior statuses and actions.

Out of extensive benchmarking, SDA outperforms the approach proposed in Habitat 3.0 (Puig et al., 2024) and a second adapted best-performing method (Cancelli et al., 2023) from Habitat

2.0 (Szot et al., 2021). We conduct a thorough experimental evaluation of the core contribution of this work—learning to infer social dynamics from (privileged) information about the person. Although our primary focus is on algorithm development, we also seek to improve the robustness of our method for potential real-world applications. To this end, we conduct experiments that bridge the gap to real-world scenarios by introducing noise to the input, reducing the refresh rate of the sensors, and modifying the simulator to reflect more realistic human behavior. Our ablative studies reveal that human trajectories are not only strong input information for the robot control policy but also provide better supervision for inferring the social dynamics latent to the same policy. Other oracular information, such as the humanoid GPS (direction and distance from the person to the robot), serves as powerful sensors for the control policy but does not facilitate adaptable social dynamics for inferring human-robot-scene interactions.

## 2 RELATED WORK

**Embodied Navigation.** Recently, there has been a surge in exploring indoor navigation within an embodied framework (Deitke et al., 2022). This upswing has been facilitated primarily by the availability of large-scale datasets comprising 3D indoor environments (Chang et al., 2017; Shen et al., 2021; Ramakrishnan et al., 2021) and simulators designed for simulating navigation within these dynamic 3D spaces (Savva et al., 2019; Shen et al., 2021; Kolve et al., 2017). Nevertheless, these simulators are not equipped to handle human entities within the environments, restricting the investigation to navigation tasks in scenarios where the agent functions independently or, at most, alongside humans simulated via static meshes that simulate movement (Xia et al., 2020; Yokoyama et al., 2022). These simulated humans are treated as dynamic obstacles and lack compliance with any social construct. This constraint has been effectively addressed with Habitat 3.0 (Puig et al., 2024), the simulator used for this research. Habitat 3.0 introduces the capability to simulate the behaviours of humans engaging in tasks within dynamic environments, thus overcoming the limitations mentioned above.

Thanks to these simulators the realm of EAI has witnessed the introduction of numerous tasks (Deitke et al., 2022), including PointGoal Navigation (Wijmans et al., 2019b), ObjectGoal Navigation (Batra et al., 2020), Embodied Question Answering (Wijmans et al., 2019a), and Vision and Language Navigation (VLN) (Anderson et al., 2018; Krantz et al., 2020). Various modular approaches (Campari et al., 2022; Chaplot et al., 2020b;a; Ramakrishnan et al., 2022; Raychaudhuri et al., 2024) have been proposed to address the challenges of navigating through static, single-agent environments. These approaches utilize maps constructed from depth images and conduct path planning directly on these maps. However, these approaches are unsuitable in social settings where dynamic objects (humans) move within the environment. This is because humans are observable only within the agent's field of view (FOV). Moreover, the agent must address the additional challenge of tracking and mapping. End-to-end RL-trained policies (Wani et al., 2020; Partsey et al., 2022; Ye et al., 2020; Campari et al., 2020; Ye et al., 2021), should be adapted to learn also social clues similarly to (Cancelli et al., 2023), where the agent learned proxemics information about the humans moving in the environments thanks to two proximity tasks. In this paper, instead, we try to learn social behaviours directly by internally modeling the humanoid trajectories in the latent representation of the agent. Furthermore, differently from (Cancelli et al., 2023), the agent's aim is no longer just avoiding collisions. Still, it involves locating this dynamically acting human and following them for a specified number of steps while maintaining a safe distance. This evolution of the task demands a heightened level of social comprehension from the agent, requiring the anticipation of the person's intentions and the ability to trail them closely without compromising safety.

**Socially-Aware Navigation.** Research in robotics, computer vision, and the analysis of human social behavior explored socially aware representations and models (Möller et al., 2021). Extending from the realm of collision-free multi-agent navigation (Berg et al., 2011; Van den Berg et al., 2008; Chen et al., 2017b; Long et al., 2017) and navigation in dynamic environments (Aoude et al., 2013), researchers have further expanded their investigations to encompass scenarios involving human presence (Guzzi et al., 2013; Ferrer et al., 2013; Chen et al., 2017a; Lu et al., 2022; Chen et al., 2019).

The approach presented in (Chen et al., 2017a) incorporates collision avoidance algorithms like CADRL (Chen et al., 2017b) while introducing common-sense social rules. This integration aims to reduce uncertainty while minimizing the risk of encountering the Freezing Robot Problem (Trautman & Krause, 2010). Other works (Ferrer et al., 2013; Chen et al., 2019) seek to model human-agent

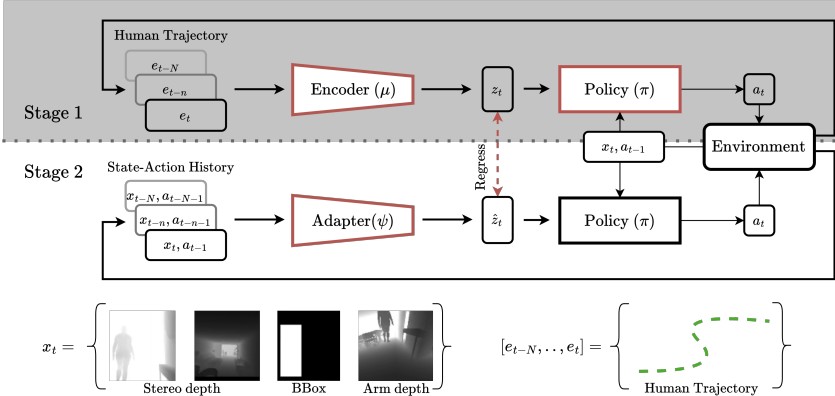

Figure 2: Pipeline of the novel methodology proposed. First, we jointly learn to encode human trajectories and a motion policy. In the next stage, given the previous states and actions, we infer the social dynamics and pass the estimated latent vector to the frozen policy.

interaction by employing techniques such as spatiotemporal graphs (Lu et al., 2022). Typically, these methods undergo testing in a minimal number of environments that offer complete knowledge about the human positions and velocities (Chen et al., 2019; 2017b), featuring simple obstacles and often assuming collaboration between moving agents(Ferrer et al., 2013; Chen et al., 2019). In contrast, our focus revolves around SocialNav within expansive indoor environments, characterized by partial knowledge about them, since the agent perceives the environment only through its sensors from an egocentric perspective, and no information about the velocity or position of the human is given to the agent. Our SDA addresses the missing information from the interacting human position, inferring it from the robot's history of actions and status.

**Modelling dynamic environments.** Simulated environments (Li et al., 2021; Tsoi et al., 2022; Puig et al., 2018) offer privileged information about the scene whose exploitation can be computationally intense or unfeasible during deployment. While navigating social environments, it is vital to take into consideration human behaviour (Kivrak et al., 2021). Ideally, one would want to forecast people's position for better path planning (Patle et al., 2019), but forecasting robot-person interactions is significantly slower (Rahman et al., 2023) than navigation policies, hence being challenging to be considered for training or deployment. To overcome this problem, we leverage literature on system identification (Ahmed & Qin, 2009; Guo et al., 2016) to infer the encoded privileged information during robot navigation. Once encoded in a latent space during the first training stage and used to train the primary policy, it is possible to asynchronously derive that same information from the state-action history (Kumar et al., 2021; Loquercio et al., 2023; Zhang et al., 2023; Liang et al., 2024; Kumar et al., 2022; Qi et al., 2023), influenced by the signal we want to identify. Unlike previous works, we are the first to identify the social dynamics (under the form of human trajectories), with the intuition that modeling human behavior is fundamental for efficient human-robot collaboration.

## 3 METHODOLOGY

This section introduces SDA: a novel framework for social navigation that incorporates human trajectories into the sequential decision-making process. The first stage of our approach focuses on encoding human trajectories into a latent social context to represent the social dynamics that are functional to the agent motion policy. In the second stage, we introduce the Adapter module, which enables the estimation of social information from the agent's past behavior. Recovering this signal allows the robot to operate without explicitly representing human behavior. We detail the training process, trajectory modeling, and optimization techniques utilized in our approach, highlighting its effectiveness in addressing the challenges of social navigation.

**Problem formalization.** The problem requires to locate and follow a humanoid in motion within an indoor environment, maintaining a distance of 1 to 2 meters for at least $k$ steps (Puig et al.,

2024). With the status $x_t$, we represent the agent's "perception" at time $t$. We exploit depth images from different cameras placed on the robot and a preprocessed version containing a humanoid detection bounding box. These perceptions are processed with a ResNet (He et al., 2016) before being fed to the recurrent policy network, selecting the best action $a_t$ at time $t$. We use Decentralized Distributed Proximal Policy Optimization (DD-PPO) (Wijmans et al., 2019b) to iteratively improve the agent's policy while maximizing rewards derived from interactions with several environments executed in parallel, ensuring stability through controlled policy updates and lower training times. The pipeline described above can be considered a baseline implementation that does not contain privileged information but relies only on the robot's onboard sensors. To address the social aspect required by the task, we consider additional details on the humanoid, e.g., "social dynamics", defined as $e_{t-N:t-1}$, where $N$ refers to the trajectory length. $e_{t-n}$ represents the position $n$ steps before time $t$, and $e_{t-N}$ refers to the earliest position in the trajectory under consideration. The notation $e_{t-N:t-1}$ is used as a shorthand to indicate the complete trajectory from $N$ steps in the past to the current time, representing absolute x-y coordinates. In the following sections, we describe how exploiting this information at training time can improve performance during deployment when it is absent.

**Stage 1: Social Policy.** Recurrent policies such as DD-PPO take as input the current status, in our case, a collection of depth images processed via a Resnet $x_t$, and the action at the previous time-step $a_{t-1}$. We add another input to this pipeline, namely a latent vector $z_t$ built by encoding the humanoid privileged information $e_{t-N:t-1}$:

$$z_t = \mu(e_{t-N:t-1}) \tag{1}$$
$$a_t = \pi(x_t, a_{t-1}, z_t) \tag{2}$$

Intuitively, by training everything with the same objective $z_t$ encodes the social dynamics that led the agent to maximize its reward, adapting to human movement patterns. Additionally, including trajectory data allows the agent to learn from past interactions and experiences. In our approach, the trajectory encoder ($\mu$) is implemented as Multilayer Perceptrons (MLPs). The objective retains its usual formulation without any explicit reference to the human trajectories:

$$L^{CLIP} = \mathbb{E}_t \left[ \min \left( r_t(\theta)\hat{A}_t, \text{clip}\left( r_t(\theta), 1-\epsilon, 1+\epsilon \right) \hat{A}_t \right) \right] \tag{3}$$

where $r_t(\theta)$ is the ratio of the probability of action $a_t$ under the current policy and the previous one that is being executed for gathering data. $\hat{A}_t$ represents the advantage function at time $t$, guiding the policy towards actions that yield higher expected rewards. Defining what information is considered "privileged" and why is essential. In our context, it refers to detailed knowledge about the humanoid in the environment, such as the exact position of humanoids defining a trajectory (traj.), or the relative position with respect to the agent often denoted as humanoid GPS (Puig et al., 2024)(hGPS). We can easily gather this information in simulated environments. However, collecting them in real-life scenarios is often impractical. This distinction is crucial as it highlights the challenge of transferring learned policies from simulation to the real world.

**Stage 2: Social Dynamics Regression.** We aim to extract and exploit social cues directly from the robot's perception and eliminate the need for auxiliary devices like GPS trackers on humanoids. Inspired by (Kumar et al., 2021), we introduce the "social dynamics" module (Adapter), parametrized by an MLP $\psi$ that takes as input the recent history of the robot's states $x_{t-N:t-1}$ and actions $a_{t-N:t-1}$ to generate a new latent vector $\hat{z}_t$:

$$\hat{z}_t = \psi(x_{t-N:t-1}, a_{t-N:t-1}) \tag{4}$$

We obtain the state-action history by deploying the agent in the environment with optimal policy $\pi^*$ obtained after the first stage and the latent vector $\hat{z}_t$:

$$a_t = \pi^*(x_t, a_{t-1}, \hat{z}_t) \tag{5}$$

During this process, we optimize the Adapter, MLP, with a supervised regression objective, Mean Squared Error (MSE), to recover the original information contained in $z_t$ that we compute relying on the preferential information trajectory, $\text{MSE}(\hat{z}_t, z_t) = \|\hat{z}_t - z_t\|_2^2$. Once we finalize the Adapter

training, instead of relying on the privileged information, we can depend upon the robot's states $x_{t-N:t-1}$ and actions $a_{t-N:t-1}$ to generate $\hat{z}_t$, serving as an estimate of the actual latent social dynamics vector $z_t$. Doing so enables the agent to estimate social dynamics online, improving its performance in dynamic environments and enhancing its social navigation capabilities, freeing it from dependence upon external sensors.

## 4 RESULTS

Section 4.1 outlines our findings on Social Navigation, along with an ablation analysis of adaptable information. Additionally, Section 4.2 offers a qualitative examination of our results, and an analysis of the role played by the latent vector $\hat{z}_t$. A more detailed analysis and further qualitative results can be found in the Appendix.

**Simulator.** We tested SDA on Habitat 3.0 (Puig et al., 2024), a simulation platform designed for human-robot interaction within domestic settings. This platform offers precise humanoid simulation capabilities with a focus on collaborative tasks such as Social Navigation and Social Rearrangement. It offers a vast library of avatars featuring multiple genders, body shapes, and appearances. Furthermore, it employs an oracle policy to generate movement and behavior, enabling programmable control of avatars for navigation, object interaction, and a range of other movements.

**Baselines.** The baselines we employ in our study are drawn from Habitat 3.0 (Puig et al., 2024) and consist of: **(i) Heuristic Expert**: a heuristic baseline equipped with access to the environment map, employing a shortest path planner to devise a route to the current location of the humanoid. The heuristic expert operates on the following principles: When the agent is beyond a distance of 1.5 meters from the humanoid, it employs a "find" behavior, utilizing a path planner to approach the humanoid. Conversely, if the humanoid is within 1.5 meters, the expert executes a backup motion to prevent collision with the humanoid. **(ii) Baseline**: the current SotA method (Puig et al., 2024), a recurrent neural network policy trained with DD-PPO (Wijmans et al., 2019b), operates on a "sensors-to-action" paradigm. Inputs to this policy consist of an egocentric arm and stereo depth sensors, a humanoid detector, and humanoid GPS coordinates, while the outputs are velocity commands in the robot's local frame. Table 1 compares our model in a realistic configuration, where the humanoid GPS data are unavailable. **(iii) Proximity tasks**: we also adapted the Proximity Tasks defined in (Cancelli et al., 2023) and applied them to the baseline (Puig et al., 2024). These tasks were proposed for a different setup of SocialNav, where the agent acts in an environment with multiple humanoids and must navigate from point A to point B while avoiding collisions. We adapted the risk and compass proximity tasks to the SocialNav setting addressed in this article. In this context, the risk has a low value (close to 0) when the agent is within 1 to 2 meters from the humanoid and a value close to 1 when the distance is less than 1 meter or greater than 2 meters. Similarly, given the presence of only one humanoid in the environment, the compass was redefined to predict the angle between the humanoid and the agent. This adjustment aims to assist during the following phase, enabling the agent to follow the humanoid while maintaining a safe distance and staying aligned with the human. The proximity tasks are jointly trained with the policy and detached during evaluation.

**Metrics.** We used the metrics for the SocialNav task as defined in (Puig et al., 2024). *Finding Success (S)* is the ratio of the episodes where the agent located and reached the human. *Finding Success Weighted by Path Steps (SPS)* measures the optimality of the path taken by the agent wrt the optimal number of steps needed to reach the human. *Following Rate (F)* is the ratio of steps during which the robot maintains a distance of 1-2 meters from the humanoid while facing towards it relatively to the maximum possible following steps. *Collision Rate (CR)* is the ratio of the episodes that ended with the robot colliding with the humanoid. *Episode Success (ES)* measures the ratio of the episodes where the agent found the human and followed it for the required number of steps, maintaining a safe distance in the 1-2 meters range.

**Privileged information.** In our work, the privileged information under consideration includes humanoid GPS (hGPS) and human trajectories (traj.). Humanoid GPS is represented in polar coordinates, a method of specifying a point's position in a plane using two parameters: the distance from the point to the origin (radius) and the angle formed between a reference direction (typically the positive x-axis) and a line connecting the origin to the point (polar angle or azimuth). In our context,

the origin is defined as the robot's position; thus, its position is implicitly known along with that of the human. Conversely, trajectories solely consist of information derived from the human's position within the environment.

## 4.1 QUANTITATIVE RESULTS

In Table 1, we present the results obtained in Habitat 3.0 (Puig et al., 2024) for the Social Navigation task. The table is divided into three sections. The first one displays outcomes achieved by utilizing a heuristic expert endowed with extensive information, including its position, map data, and the humanoid's position, granting it a competitive advantage over other methods. The subsequent section features models trained and tested using ground truth (GT) data (Baselines and Stage 1), thus establishing an upper limit for techniques utilizing more practical inputs feasible in real-world scenarios. Lastly, the final rows delineate results from methods conducting inference without privileged information. The models that use privileged information (GT) show that S1 has lower performance than the two Baselines, especially in $S$ and $SPS$. This is to be expected, considering that while trajectories solely depict the movement of the human in the environment, hGPS furnishes crucial details on locating the humanoid, offering insights into the distance and angle between them. In the second stage, despite the absence of human trajectory input, SDA keeps the performance level of Stage 1 by adapting to the social dynamics. Our model outperforms the baselines in the find task, increasing $S$ and $SPS$ by 6%.

Our approach generally improves performance in the episode success ($ES$) metric, which occurs when the agent finds the humanoid and follows it for 400 steps. However, we emphasize that the test-time episode ends at 1500 steps or when the agent collides with the humanoid (Puig et al., 2024), not after the 400 follow steps. In this context, $S$, $SPS$, and $F$ metrics demonstrate how SDA, compared to the Baseline, more frequently locates the humanoid, follows a more optimal path (on average 438 vs. 540 steps), and follows it for a longer duration (390 vs. 218 steps). Therefore, as the agent follows the humanoid longer, it has more chances to collide with it, given that the episode does not necessarily end after the required steps. In a scenario where the test-time episode concludes either after 400 follow steps or immediately upon a collision between the agent and the humanoid, SDA and the Baseline show a comparable collision rate ($CR$), 0.39, and 0.38, respectively.

Table 1: Main results for Social Navigation. Within the table, GT denotes ground truth privileged information and * corresponds to reproduced results.

| Models | hGPS | traj. | S ↑ | SPS ↑ | F ↑ | CR ↓ | ES ↑ |
|---|---|---|---|---|---|---|---|
| Heuristic Expert Puig et al. (2024) | - | - | 1.00 | 0.97 | 0.51 | 0.52 | - |
| Baseline Puig et al. (2024) | GT | | $0.97^{\pm0.00}$ | $0.65^{\pm0.00}$ | $0.44^{\pm0.01}$ | $0.51^{\pm0.03}$ | $0.55^{\pm0.01}*$ |
| Baseline+Proximity Cancelli et al. (2023)[1] | GT | | $0.97^{\pm0.01}$ | $0.64^{\pm0.00}$ | $0.57^{\pm0.01}$ | $0.58^{\pm0.03}$ | $0.63^{\pm0.02}$ |
| SDA - S1 | | GT | $0.92^{\pm0.00}$ | $0.46^{\pm0.01}$ | $0.44^{\pm0.02}$ | $0.61^{\pm0.02}$ | $0.50^{\pm0.01}$ |
| Baseline Puig et al. (2024) | | | $0.76^{\pm0.02}$ | $0.34^{\pm0.01}$ | $0.29^{\pm0.01}$ | $\mathbf{0.48}^{\pm0.03}$ | $0.40^{\pm0.02}*$ |
| Baseline+Proximity Cancelli et al. (2023) | | | $0.85^{\pm0.02}$ | $0.41^{\pm0.02}$ | $0.37^{\pm0.01}$ | $0.58^{\pm0.02}$ | $0.41^{\pm0.01}$ |
| SDA - S2 | | | $\mathbf{0.91}^{\pm0.01}$ | $\mathbf{0.45}^{\pm0.01}$ | $\mathbf{0.39}^{\pm0.01}$ | $0.57^{\pm0.02}$ | $\mathbf{0.43}^{\pm0.02}$ |

**Adaptable information.** Table 2 presents the results from Stage 1 (S1) and Stage 2 (S2) with the utilization of various privileged information, such as Humanoid GPS (hGPS) and human trajectories (traj.). During S1, particularly in $S$ and notably in $SPS$, incorporating hGPS as an additional input leads to superior results. This advantage is likely attributed to hGPS containing implicit information about human and robot positions, facilitating more efficient path selection, especially in $SPS$ scenarios. However, this pattern is not evident in S2, where trajectories typically provide more adaptable information. As previously mentioned, hGPS inherently encompasses the robot's position, posing challenges in regressing this data during the second stage due to the lack of initial context regarding the robot's location relative to the environment. Furthermore, hGPS may be difficult to adapt when the human is detected and disappears around a wall. Since hGPS comprises polar coordinates, the distance between the robot and the human spans the wall. This issue does not arise when utilizing trajectories, as they do not require any information about the robot and can be adapted solely based on depth images and detection information.

---
[1]Code refactored and adapted from Cancelli et al. (2023) to Habitat 3.0.

Table 2: Ablation studies for social dynamics estimation. GT denotes ground truth privileged information, and $_A$ indicates the adapted information.

| Models | hGPS | traj. | hGPS$_A$ | traj.$_A$ | S ↑ | SPS ↑ | F ↑ | CR ↓ | ES ↑ |
|---|---|---|---|---|---|---|---|---|---|
| S1 | GT | GT | | | $\mathbf{0.94}^{\pm 0.01}$ | $0.58^{\pm 0.00}$ | $0.45^{\pm 0.02}$ | $0.64^{\pm 0.03}$ | $\mathbf{0.52}^{\pm 0.01}$ |
| S1 | GT | | | | $0.93^{\pm 0.00}$ | $\mathbf{0.62}^{\pm 0.01}$ | $\mathbf{0.46}^{\pm 0.01}$ | $0.64^{\pm 0.02}$ | $0.48^{\pm 0.01}$ |
| S1 (*Proposed*) | | GT | | | $0.92^{\pm 0.00}$ | $0.46^{\pm 0.01}$ | $0.44^{\pm 0.02}$ | $\mathbf{0.61}^{\pm 0.02}$ | $0.50^{\pm 0.01}$ |
| S2 | | | ✓ | ✓ | $0.57^{\pm 0.06}$ | $0.21^{\pm 0.04}$ | $0.05^{\pm 0.01}$ | $\mathbf{0.30}^{\pm 0.02}$ | $0.02^{\pm 0.00}$ |
| S2 | | | ✓ | | $0.70^{\pm 0.02}$ | $0.31^{\pm 0.02}$ | $0.05^{\pm 0.01}$ | $0.70^{\pm 0.03}$ | $0.03^{\pm 0.01}$ |
| S2 (*Proposed*) | | | | ✓ | $\mathbf{0.91}^{\pm 0.01}$ | $\mathbf{0.45}^{\pm 0.01}$ | $\mathbf{0.39}^{\pm 0.01}$ | $0.57^{\pm 0.02}$ | $\mathbf{0.43}^{\pm 0.02}$ |

## 4.2 QUALITATIVE RESULTS

We qualitatively demonstrate the results in our proposed SDA. Firstly, we showcase the agent's ability to locate the humanoid within the environment by moving around, followed by its capability to follow the humanoid within the environment. Subsequently, we present two specific behaviors where the agent briefly spots the humanoid and one where it moves backward to create space for passage. Fig. 3 shows an episode where the agent and the human are located in different rooms at the start. Then, the agent begins its search for the humanoid by navigating within the environment until the encounter takes place. After the encounter, the agent then transitions into the follow phase.

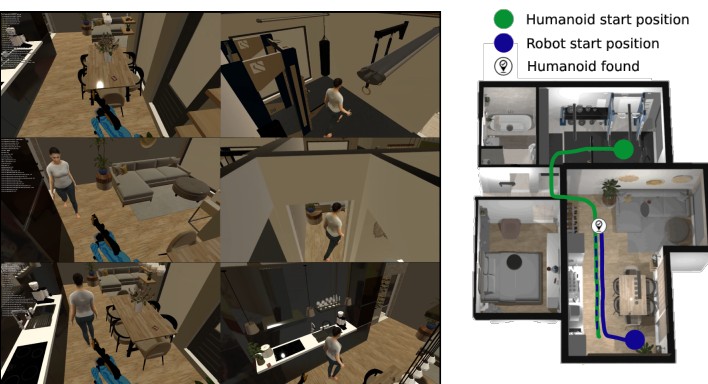

Figure 3: The agent and the humanoid start the episode in separate rooms. The agent navigates through the environment in search of the humanoid, and once found, begins to follow it.

**Latent Analysis.** We further investigate the implications of our approach and the role of inferred human behaviors in robot decision-making. In Figure 4, we present a latent space analysis of behaviors, where the t-SNE projection reveals four key behavioral stages. **Find**: The robot has not yet detected a human in its RGB camera frames and remains outside the zone of interest, typically at a distance greater than 1-2 meters. **Seek**: The human has been detected but is still beyond the zone of interest. The robot moves toward the human to reduce the distance. **Lost**: The robot has lost sight of the human it previously detected. However, the person remains within the 1-2 meter range, prompting the robot to reorient and attempt reacquisition. **Follow**: The human is actively tracked and remains within the 1-2 meter zone, allowing the robot to continue following the person. On the left, dots overlaid with a gradient from black to white illustrate a representative experiment, where the color gradient indicates the progression of time. This visualization effectively captures the robot's adaptive behavior as it transitions between modes: the robot starts in **Find** mode, shifts to **Seek**, and then enters **Follow** mode upon detecting the person. At times, when the human moves behind an obstacle (e.g., a wall), the robot switches to **Lost** mode. Eventually, the robot reacquires the person and resumes following until the episode concludes. The cluster-colored scatter plot on the right further emphasizes the overlap between behaviors like **Seek** and **Follow**, as both involve observing and responding to the human's position, leading to shared characteristics in the latent space. Similarly, the transitions between stages such as **Lost** and **Seek** are continuous rather than discrete, aligning with the fluid nature of human-robot interactions.

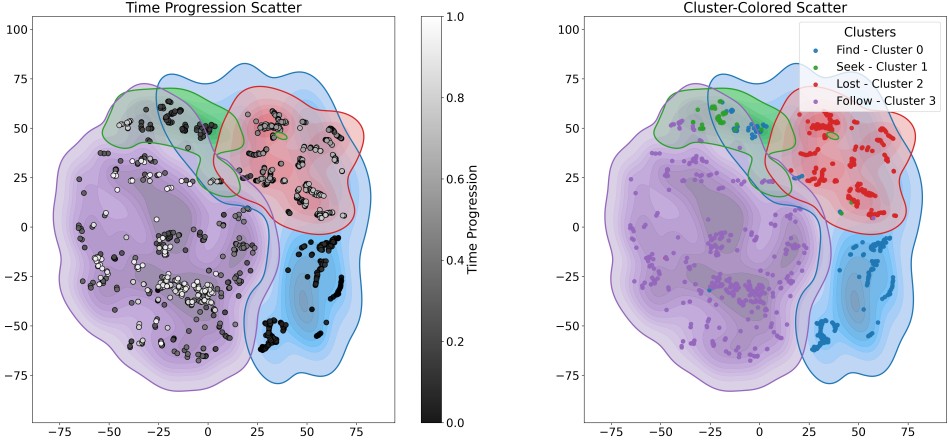

Figure 4: Latent Analysis

## 5 TOWARD REAL-WORLD SCENARIOS

While simulated environments have advanced significantly, they still fall short in capturing the full unpredictability of real-world interactions. To address this limitation, we take incremental steps toward real-world scenarios by proposing tests that incorporate more realistic social behavior, constrained computational resources, imperfect communication between the adaptation module and the primary policy, and noisy sensors. Although conducted in simulation, our work aligns with standard practices in the Embodied AI literature (Campari et al., 2020; Raychaudhuri et al., 2024; Cancelli et al., 2023; Chaplot et al., 2020a;b; Ye et al., 2020; 2021; Yokoyama et al., 2022; Majumdar et al., 2024), which utilize simulated environments to rigorously evaluate and iterate on agent behaviors. In contrast, "Sim2Real" studies are typically standalone works (Gervet et al., 2023; Partsey et al., 2022) that focus specifically on transferring policies and learned behaviors from simulated environments to real-world applications. The anticipated conclusion is that more realistic human behavior and sensor readings likely contribute to the sim-to-real gaps. However, incorporating realistic training samples can partially recover performance loss. Additionally, the performance gap identified between the Baseline (Puig et al., 2024) and our proposed SDA remains consistent across metrics, underscoring the algorithmic novelty of our simulated-only study.

**ORCA.** We have augmented the motion policy of humanoids in Habitat 3.0 by making them aware of the robot presence, with reciprocal collision avoidance (ORCA) (Snape et al., 2010). When used on two agents (or more), it provides sufficient conditions for collision-free motion by letting each agent take half the responsibility of avoiding pairwise collisions. In our case, however, it is applied only to the humanoid; meanwhile, the robot still relies on its end-to-end policy. This removes unrealistic behaviors where the human sees the robot and goes straight into it. Table 3 shows a lower collision rate (CR), 37% instead of 57% and a higher episode success (ES) 48% vs. 43%; meanwhile, we notice a slight decrease in performance in the other metrics.

Table 3: Comparison of SDA performances on plain Habitat 3.0 versus the variant with ORCA.

| SDA | S ↑ | SPS ↑ | F ↑ | CR ↓ | ES ↑ |
|---|---|---|---|---|---|
| Habitat 3.0 | $\mathbf{0.91}^{\pm 0.01}$ | $\mathbf{0.45}^{\pm 0.01}$ | $\mathbf{0.39}^{\pm 0.01}$ | $0.57^{\pm 0.02}$ | $0.43^{\pm 0.02}$ |
| Habitat 3.0 + ORCA | $0.90^{\pm 0.01}$ | $0.43^{\pm 0.02}$ | $0.38^{\pm 0.01}$ | $\mathbf{0.37}^{\pm 0.01}$ | $\mathbf{0.48}^{\pm 0.01}$ |

**Lower frequency updates.** In Table 4, we simulate a scenario where the adaptation module operates at a lower frequency (i.e., with a reduced update rate) due to potential computational constraints during deployment. In SDA, the $\hat{z}$ vector is typically updated at each timestep. We tested two alternative settings where updates occur once every two or every one hundred timesteps. Interestingly, this results in only a slight performance degradation in the metrics related to the finding aspect of the task, while overall performance improves (ES), driven by enhancements in the following (F) task

itself. This improvement can be attributed to slower sensor update rates encouraging the agent to focus on more stable behaviour avoiding unnecessary short-term adjustments, leading to smoother navigation and more effective long-term strategies.

Table 4: SDA performance considering missing readers on Habitat 3.0.

| Update Rate | S ↑ | SPS ↑ | F ↑ | CR ↓ | ES ↑ |
|---|---|---|---|---|---|
| 1 *(Proposed)* | **0.91**$^{\pm0.01}$ | **0.45**$^{\pm0.01}$ | 0.39$^{\pm0.01}$ | **0.57**$^{\pm0.02}$ | 0.43$^{\pm0.02}$ |
| 1/2 | 0.87$^{\pm0.01}$ | 0.39$^{\pm0.01}$ | **0.44**$^{\pm0.01}$ | 0.63$^{\pm0.02}$ | **0.48**$^{\pm0.02}$ |
| 1/100 | 0.85$^{\pm0.01}$ | 0.38$^{\pm0.01}$ | 0.43$^{\pm0.01}$ | 0.64$^{\pm0.03}$ | 0.46$^{\pm0.01}$ |

**Noisy inputs.** We evaluate the addition of noise on both the sensor input (depth images and bounding boxes) and actuators. Table 5 presents the results after fine-tuning both SDA and Baseline policies for 1M steps. We used Gaussian noise on the Bounding box human-detector, Redwood noise on the Depth camera (the policy does not use the RGB) (Partsey et al., 2022; Choi et al., 2015), and Gaussian noise on the high-level actuators (Partsey et al., 2022; Choi et al., 2015), the agent's angle and velocity. Analyzing the results in Table 5, we note that the overall largest drop in performance regards Finding Success (S), dropping from 91% to 81-83%. As a consequence of this, the collision rate (CR) actually improves since the robot needs to follow the humanoid for less time. The performance in follow (F) is mostly affected by noise in the bounding boxes (from 39% to 30%). The performance in episode success is mostly affected by adding both depth camera and bounding box noise (from 43% to 25%). Importantly, the gap between SDA and the Baseline (Puig et al., 2024) remains consistent across all noise types and all metrics, which supports the validity of tests in simulated environments.

Table 5: Ablation study with RedWood Noise on the Depth Camera, and Gaussian Noise on the Bounding Box and Actuators.

| Model | Noisy Input | S ↑ | SPS ↑ | F ↑ | CR ↓ | ES ↑ |
|---|---|---|---|---|---|---|
| Baseline | None | 0.76$^{\pm0.02}$ | 0.34$^{\pm0.01}$ | 0.29$^{\pm0.01}$ | 0.48$^{\pm0.03}$ | 0.40$^{\pm0.02}$ |
| SDA | None | 0.91$^{\pm0.01}$ | 0.45$^{\pm0.01}$ | 0.39$^{\pm0.01}$ | 0.57$^{\pm0.02}$ | 0.43$^{\pm0.02}$ |
| Baseline | Depth Camera | 0.70$^{\pm0.01}$ | 0.32$^{\pm0.01}$ | 0.26$^{\pm0.01}$ | 0.43$^{\pm0.02}$ | 0.20$^{\pm0.02}$ |
| SDA | Depth Camera | 0.83$^{\pm0.02}$ | 0.42$^{\pm0.03}$ | 0.34$^{\pm0.02}$ | 0.30$^{\pm0.01}$ | 0.37$^{\pm0.01}$ |
| Baseline | Bounding Box | 0.73$^{\pm0.01}$ | 0.30$^{\pm0.01}$ | 0.24$^{\pm0.01}$ | 0.44$^{\pm0.02}$ | 0.15$^{\pm0.02}$ |
| SDA | Bounding Box | 0.83$^{\pm0.01}$ | 0.41$^{\pm0.02}$ | 0.30$^{\pm0.01}$ | 0.28$^{\pm0.02}$ | 0.35$^{\pm0.02}$ |
| Baseline | Depth Camera + Bounding Box | 0.69$^{\pm0.01}$ | 0.33$^{\pm0.01}$ | 0.25$^{\pm0.01}$ | 0.44$^{\pm0.02}$ | 0.21$^{\pm0.02}$ |
| SDA | Depth Camera + Bounding Box | 0.82$^{\pm0.01}$ | 0.41$^{\pm0.02}$ | 0.48$^{\pm0.02}$ | 0.45$^{\pm0.01}$ | 0.25$^{\pm0.02}$ |
| Baseline | Actuators | 0.71$^{\pm0.01}$ | 0.29$^{\pm0.01}$ | 0.24$^{\pm0.01}$ | 0.42$^{\pm0.02}$ | 0.31$^{\pm0.02}$ |
| SDA | Actuators | 0.81$^{\pm0.01}$ | 0.41$^{\pm0.02}$ | 0.35$^{\pm0.01}$ | 0.42$^{\pm0.01}$ | 0.40$^{\pm0.02}$ |

**Limitations.** Table 2 illustrates how the performance in stage 2 is affected by the adaptability of information, underscoring its importance in evaluating the model's effectiveness. Additionally, a limitation of our approach is that the proposed model has been benchmarked solely in simulation, relying on training information—such as trajectories—that may not be readily available in real-world settings.

# 6 CONCLUSIONS

Our study presents the Social Dynamics Adaptation model (SDA) for Social Navigation. Notably, it is the first to integrate privileged human dynamics information during training while adapting it in the following stage, enabling its application in realistic environments without relying on such privileged information. Our findings underscore the non-trivial nature of adapting information, highlighting the necessity for selective processes. In future research, we aim to extend our model by incorporating diverse human dynamics beyond trajectories, enhancing the robot's comprehension of human movement patterns.

**Acknowledgements.** We are grateful to Panasonic for partially supporting this work. We acknowledge the financial support from the PNRR MUR project PE0000013-FAIR and from the Sapienza grant RG123188B3EF6A80 (CENTS). Thanks CINECA and the ISCRA initiative for high-performance computing resources and support.

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

# A APPENDIX

The content is structured as follows:

In this appendix, we provide additional information and detailed analyses that complement the main paper. The appendix is organized as follows:

- **Metrics (Sec. A.1):** This section expands on the metrics presented in the primary paper and introduces supplementary ones, as exemplified in (Puig et al., 2024). We explain the definitions, motivations, and computation details for each metric used in our evaluation.

- **Training Details (Sec. A.2):** Here, we outline the training methodology, including the training stages, the use of DD-PPO, and the hardware setup for training. This section provides further context on how the models were optimized.

- **Results (Sec. A.3):** This section includes comprehensive tables listing all performance metrics along with an accompanying explanation of the results. We detail how different experimental conditions affect performance.

- **Simulated Changing Human Motion Patterns (Sec. A.4):** This section details experiments where we simulate varying human motion patterns to account for scenarios with abrupt speed changes (e.g., constant, reduced, and random speeds). We compare the performance of SDA under these conditions.

- **Statistical Significance and Robustness (Sec. A.5):** In this section, we present statistical analyses (Welch's t-test) to validate the improvements of SDA over the baseline and Cancelli et al. (Cancelli et al., 2023). The reported t-statistics and p-values confirm that the performance enhancements are statistically significant.

- **Adaptability to Multi-Human Interactions (Sec. A.6):** This section evaluates the performance of SDA in environments involving one, two, and three humans. We show how the increased task complexity affects key metrics such as success rate, following frequency, and collision rate, and demonstrate that SDA consistently outperforms the baseline.

- **Error Analysis (Sec. A.7):** We analyze failure cases, identifying key areas where the model encounters challenges such as constrained movements and blind spots. The analysis is supported by quantitative and qualitative examples.

- **Ablation Studies (Sec. A.8):** This section explores the effect of privileged information and trajectory length on model performance. We present ablation experiments that isolate these factors.

- **Qualitative Results (Sec. A.9):** Visual examples of episodes where the agent successfully tracks or avoids collisions with humans are presented here, accompanied by qualitative analysis to illustrate strengths and failure modes.

Moreover, we show some generated episodes in the enclosed supplementary video.

## A.1 METRICS

In our main paper, we utilize metrics including Finding Success ($S$), Finding Success Weighted by Path Steps ($SPS$), Following Rate ($F$), Collision Rate ($CR$), and Episode Success ($ES$). However, Habitat 3.0 (Puig et al., 2024) introduces in their supplementary material additional metrics such as Backup-Yield Rate ($BYR$), Total Distance ($TD$), and Following Distance ($FD$), providing further insights into the models. Similarly, in our supplementary material, we also incorporate these metrics. Subsequently, a comprehensive explanation will be given for all the former and latter metrics featured.

(1) **Finding Success** ($S$): This metric, denoted $S$, evaluates whether the robot successfully locates the humanoid within the maximum episode steps and reaches it within a close range of 1-2 meters while facing toward it. It is represented as:

$$S = \begin{cases} 1 & \text{if the robot successfully finds and reaches the humanoid,} \\ 0 & \text{otherwise} \end{cases}$$

This metric provides a binary indication of the robot's ability to locate and approach the humanoid within the designated constraints.

(2) **Finding Success Weighted by Path Steps** ($SPS$): The $SPS$ metric, calculated as $SPS = S \cdot \frac{l}{\max(l,p)}$, evaluates the efficiency of the robot's path relative to an oracle with complete knowledge of the humanoid's trajectory and the environment map. Here, $l$ represents the minimum steps an oracle would take to find the humanoid, and $p$ denotes the agent's actual path steps. A higher $SPS$ value indicates the robot's more efficient path toward finding the humanoid.

(3) **Following Rate** ($F$): The following rate $F$ quantifies the ratio of steps during which the robot maintains a distance of 1-2 meters from the humanoid while facing towards it relative to the maximum possible following steps. It is calculated as:

$$F = \frac{w}{\max(E - l, w)}$$

$E$ denotes the maximum episode duration, and $w$ represents the number of steps during which the agent closely follows the humanoid. This metric provides insight into the robot's ability to consistently track the humanoid once it has been located.

(4) **Collision Rate** ($CR$): The collision rate $CR$ measures the ratio of episodes that end with the robot colliding with the humanoid. It is computed as:

$$CR = \frac{\text{Number of episodes ending in collision}}{\text{Total number of episodes}}$$

This metric assesses the robot's collision avoidance capabilities during interactions with the humanoid.

(5) **Backup-Yield Rate** ($BYR$): The backup-yield rate $BYR$ quantifies the frequency with which the robot performs backup or yield motions to avoid collision when the humanoid is nearby. A 'backup motion' refers to a backward movement executed by the robot when the distance between the robot and the humanoid is less than 1.5 meters. Similarly, a 'yield motion' denotes a robot's maneuver to avoid collision when the distance between them is less than 1.5 meters and the robot's velocity is less than 0.1 m/s. The $BYR$ is computed as:

$$BYR = \frac{\text{Number of backup or yield motions}}{\text{Total number of episodes}}$$

This metric provides insights into the effectiveness of the robot's collision avoidance strategies.

(6) **Total Distance** between the robot and the humanoid ($TD$): The $TD$ metric evaluates the average L2 distance between the robot and the humanoid over the total number of episode steps. It is calculated as:

$$TD = \frac{\sum \text{L2 distance between robot and humanoid}}{\text{Total number of episode steps}}$$

This metric measures the overall proximity between the robot and the humanoid throughout the episodes, providing insights into the effectiveness of the robot's navigation and tracking capabilities.

(7) **Following Distance** between the robot and the humanoid after the first encounter ($FD$): The $FD$ metric assesses the L2 distance between the robot and the humanoid after the robot initially encounters the humanoid. It quantifies the proximity between the two entities during the following stages of interaction. The $FD$ should ideally be maintained within 1-2 meters, indicating effective tracking and following behavior. This metric is calculated as:

$$FD = \frac{\sum \text{L2 distance between robot and humanoid after first encounter}}{\text{Total number of episode steps}}$$

The $FD$ metric provides valuable insights into the robot's ability to maintain an appropriate distance from the humanoid target after initial contact, which is crucial for effective interaction and task completion.

(8) **Episode Success** ($ES$) measures the ratio of the episodes where the agent found the human and followed it for the required number of steps, maintaining a safe distance in the 1-2 meters range.

A.2 TRAINING DETAILS.

During training, we encode social dynamics using a ResNet, trained from scratch. The trajectory encoder $\mu$ consists of a 3-layer MLP, and the output $z_t$ has a dimensionality of 128. The Adapter $\psi$

comprises alternating spatial and temporal MLP layers, with the output $\hat{z}_t$ matching the dimensionality of $z_t$. We jointly train the policy $\pi$, the perceptions encoder (ResNet), and the social dynamics (Trajectory Encoder) encoder $\mu$ during the first stage. In the second stage, everything is frozen except the Adapter. In Stage 1, we utilize DD-PPO (Wijmans et al., 2019b) for 250 million steps across 24 environments, following the training protocol presented in (Puig et al., 2024). Furthemore, each time step is approximately $0.04$ seconds, so we consider a trajectory of $0.8$ seconds. An entire episode of $1500$ steps corresponds to almost a 1-minute long video. Inputs to the policy are egocentric arm depth and a humanoid detector, and outputs are velocity commands (linear and angular) in the robot's local frame. This policy does not have access to a map of the environment. The training process takes approximately four days. In Stage 2, we employ supervised learning for 5 million steps across the same environments. The learning process lasts around 2 hours. Both stages utilize 4 A100 GPUs for efficient computation.

## A.3 RESULTS

Table 6: Comparative evaluation of Social Navigation on Habitat 3.0 Puig et al. (2024). Within the table, GT denotes ground truth privileged information and * corresponds to reproduced results. Beyond what is reported in Table 1 of the main paper, we additionally report here: (1) Backup-Yield Rate (BYR), (2) The Total Distance between the robot and the humanoid (TD), and (3) The "Following" Distance (in *meters*) between the robot and the humanoid after the first encounter (FD).

| Models | hGPS | traj. | S ↑ | SPS ↑ | F ↑ | CR ↓ | BYR | TD | FD | ES ↑ |
|---|---|---|---|---|---|---|---|---|---|---|
| Heuristic Expert Puig et al. (2024) | - | - | 1.00 | 0.97 | 0.51 | 0.52 | 0.24 | 2.56 | 1.72 | - |
| Baseline Puig et al. (2024) | GT | | 0.97 | 0.65 | 0.44 | 0.51 | 0.19 | 3.43 | 1.70 | 0.55* |
| Baseline+Prox. Cancelli et al. (2023)[2] | GT | | 0.97 | 0.64 | 0.57 | 0.58 | 0.17 | 3.15 | 1.66 | 0.63 |
| SDA - S1 | | GT | 0.92 | 0.46 | 0.44 | 0.61 | 0.18 | 3.70 | 1.83 | 0.50 |
| Baseline Puig et al. (2024) | | | 0.76 | 0.34 | 0.29 | **0.48** | 0.13 | 5.18 | 1.64 | 0.42* |
| Baseline+Prox. Cancelli et al. (2023) | | | 0.85 | 0.41 | 0.37 | 0.58 | 0.14 | 4.24 | 1.57 | 0.41 |
| SDA - S2 | | | **0.91** | **0.45** | **0.39** | 0.57 | 0.12 | 4.24 | 1.80 | **0.43** |

In Table 6, we compare Baseline (Puig et al., 2024), Baseline+Prox. (Cancelli et al., 2023), and the novel SDA model on the additional metrics proposed by (Puig et al., 2024). We showcase a comparable *Backup Yield Rate* across the methodologies, meaning that all models suggest an avoidance behaviour. Regarding *Following Distance*, all models, on average, fall within the ideal 1-2 meters range, with SDA exhibiting a conservative behavior, i.e., SDA remains at 1.80 meters further from the humanoid. The *Total Distance* is lower for SDA and Baseline+Prox. than for Baseline. This is due to the ability of the first two models to detect the human ($SPS$) earlier and to follow it for a longer duration ($F$). In summary, the inclusion of additional metrics such as $BYR$, $TD$, and $FD$ leads to analogous conclusions, as previously outlined in the main paper (Sec. 4.1). Specifically, it underscores that the comprehensive advantage in adapting to social dynamics stems from following a more optimal trajectory, maintaining it over an extended period, and ensuring a safe distance.

## A.4 SIMULATING CHANGING HUMAN MOTION PATTERNS

We simulate varying motion patterns of humans to account for scenarios where individuals may abruptly slow down due to impediments or reduced mobility. These experiments complement our ablation studies on sensor noise and processing latency (see Section 5). *Note: all evaluations are based on a policy trained with constant human speed.*

Table 7 compares the performance of SDA Stage 2 under three different human motion settings:

- **constant:** Humans move at a fixed speed.
- **constant/2:** Humans move at half the constant speed, simulating reduced mobility.
- **random:** Human speed varies randomly to emulate unpredictable behavior.

The results indicate that:

---

[2]Code refactored and adapted from Cancelli et al. (2023) to Habitat 3.0.

Table 7: Comparison of SDA Stage 2 performances using constant human speed (h.speed) or random human speed in the simulator.

| Model | h.speed | S ↑ | SPS ↑ | F ↑ | CR ↓ | ES ↑ |
|---|---|---|---|---|---|---|
| SDA - S2 | constant | $0.91 \pm 0.01$ | $0.45 \pm 0.01$ | $0.39 \pm 0.01$ | $0.57 \pm 0.02$ | $0.43 \pm 0.02$ |
| SDA - S2 | constant/2 | $0.87 \pm 0.01$ | $0.57 \pm 0.02$ | $0.28 \pm 0.01$ | $0.13 \pm 0.01$ | $0.70 \pm 0.01$ |
| SDA - S2 | random | $0.90 \pm 0.01$ | $0.45 \pm 0.02$ | $0.25 \pm 0.01$ | $0.48 \pm 0.01$ | $0.40 \pm 0.01$ |

- **Consistency in Navigation Efficiency:** Under both constant and random speeds, the Finding Success (S) and SPS scores remain comparable (0.91 vs. 0.90 and 0.45, respectively), demonstrating robust navigation efficiency.

- **Improved Safety with Reduced Speed:** When humans move at half the constant speed, the agent achieves a Finding Success (S) score of 0.87 along with a significantly lower Collision Rate (CR of 0.13), suggesting enhanced safety in slower environments.

- **Trade-off with Unpredictability:** In the random speed scenario, while the Following Rate (F) drops from 0.39 to 0.25—highlighting challenges in adapting to abrupt motion changes—the Collision Rate also improves (decreases from 0.57 to 0.48), reflecting a more cautious navigational strategy.

## A.5 STATISTICAL SIGNIFICANCE AND ROBUSTNESS OF SDA

The proposed SDA method extends the baseline Puig et al. (2024) (and not Cancelli et al. Cancelli et al. (2023)) consistently across all metrics. As shown in Table 6 of the paper, SDA improves the primary tasks of:

- **Finding the human (Finding Success, S):** Improved from 76% to 91% (an increase of 15 percentage points),

- **Reaching the human using an optimal path (SPS):** Improved from 0.34 to 0.45 (11 percentage points), and

- **Following the human (F):** Improved from 0.29 to 0.39 (10 percentage points).

While the overall improvement in Episode Success (ES) is more modest (from 0.41 to 0.43), this is primarily attributable to an increased Collision Rate (CR). As discussed in line 347 of the main paper, SDA finds the human earlier and follows them for significantly longer—on average, 390 steps instead of 218. This extended following duration introduces a greater challenge in avoiding collisions, thereby moderating the improvement observed in ES.

To evaluate the statistical significance of these improvements, we performed independent two-sample t-tests assuming unequal variances (Welch's t-test). The results, presented in Table 8, confirm that the improvements of SDA over both the baseline and Cancelli et al. Cancelli et al. (2023) are statistically significant.

Table 8: Updated t-statistics and p-values for the comparison of methods.

| Metric | Baseline+Proximity | SDA - S2 | t-statistic | p-value | Significance |
|---|---|---|---|---|---|
| S | $0.85 \pm 0.02$ | $0.91 \pm 0.01$ | -8.49 | 0.00000103 | Significant |
| SPS | $0.41 \pm 0.02$ | $0.45 \pm 0.01$ | -5.66 | 0.0000732 | Significant |
| F | $0.37 \pm 0.01$ | $0.39 \pm 0.01$ | -4.47 | 0.000295 | Significant |
| CR | $0.58 \pm 0.02$ | $0.54 \pm 0.02$ | 4.47 | 0.000295 | Significant |
| ES | $0.42 \pm 0.01$ | $0.45 \pm 0.02$ | -4.24 | 0.000924 | Significant |

## A.6 ADAPTABILITY TO MULTI-HUMAN INTERACTIONS AND TASK COMPLEXITY

While our current focus is on single-human interactions, our framework is inherently flexible and capable of adapting to more complex social settings involving multiple humans. To provide insights into performance under such conditions, we evaluated both the SDA method and a baseline in

environments containing one, two, and three humans. Note that, due to the time constraints of this rebuttal, the evaluations are based on a policy trained exclusively with a single human.

The results, summarized in Table 9, highlight the growing difficulty of the task as the number of humans increases. For example, in single-human scenarios, SDA achieves a success rate (S) of 0.91; however, this decreases to 0.70 in three-human scenarios. Similarly, the frequency of reaching the target (F) declines from 0.39 to 0.12, and the collision rate (CR) rises from 0.57 to 0.78, reflecting the increased complexity of navigating multiple dynamic obstacles while maintaining focus on the target human.

Despite these challenges, our method consistently outperforms the baseline across all metrics and scenarios. In single-human settings, SDA outperforms the baseline by 0.15 in S and 0.10 in F. In three-human environments, SDA maintains a higher success rate (0.70 vs. 0.67) and a greater frequency of reaching the target (0.12 vs. 0.07). These findings indicate that while task performance degrades with increased complexity, the SDA framework remains robust and adaptable, and the observed degradation is attributable to the inherent task difficulty rather than a limitation of our method.

Table 9: Comparison of SDA performances on Habitat 3.0 with single and multiple human scenarios.

|  | S ↑ | SPS ↑ | F ↑ | CR ↓ | ES ↑ |
|---|---|---|---|---|---|
| SDA - S2 (single-human) | $0.91 \pm 0.01$ | $0.45 \pm 0.01$ | $0.39 \pm 0.01$ | $0.57 \pm 0.02$ | $0.43 \pm 0.02$ |
| Baseline (single-human) | $0.76 \pm 0.02$ | $0.34 \pm 0.01$ | $0.29 \pm 0.01$ | $0.48 \pm 0.03$ | $0.40 \pm 0.02$ |
| SDA - S2 (two-humans) | $0.80 \pm 0.01$ | $0.41 \pm 0.02$ | $0.18 \pm 0.01$ | $0.68 \pm 0.01$ | $0.20 \pm 0.01$ |
| Baseline (two-humans) | $0.76 \pm 0.01$ | $0.36 \pm 0.02$ | $0.09 \pm 0.01$ | $0.80 \pm 0.01$ | $0.11 \pm 0.01$ |
| SDA - S2 (three-humans) | $0.70 \pm 0.01$ | $0.38 \pm 0.02$ | $0.12 \pm 0.01$ | $0.78 \pm 0.01$ | $0.15 \pm 0.01$ |
| Baseline (three-humans) | $0.67 \pm 0.01$ | $0.36 \pm 0.02$ | $0.07 \pm 0.01$ | $0.83 \pm 0.01$ | $0.09 \pm 0.01$ |

### A.7 ERROR ANALYSIS

We include in Fig.5 We analyze the distribution of failure causes in social navigation analyzed by watching 100 failed episodes and categorizing the failures into five types:

- Constrained Movements (28%): The robot cannot avoid collision due to environmental constraints.
- Blindspot (25%): The robot cannot perceive the human due to blind corners or side collisions.
- Not Found (22%): The robot does not detect the humanoid within 1500 simulation steps.
- Moving Backwards (22%): The robot yield space backwards but fails to avoid collisions.
- Walking into a Humanoid (3%): Frontal collisions with the humanoid are the least common cause.

The failure analysis offers valuable insights into areas needing improvement. Walking directly into the humanoid is very unlikely and represent the worst scenario. As can be seen from Table 3, by including ORCA, we reduce the collision rate by 20 percentage points and this scenario will likely disappear. With ORCA, humans can either slow down or slightly change their direction, particularly for Constrained Movements and moving Backwards. Constrained Movements and Not Found could be improved by high-level planning capabilities that devise better exploration strategies to locate the person and let the robot back off until there is enough space to let the human pass. The addition of a microphone sensor could also help locate humans in the scenario in which someone could call their robot.

### A.8 ADDITIONAL ANALYSIS

In the next section, we conduct a failure case analysis of SDA , examining the adaptability of privileged information and performing an ablation study on its design.

Fig. 6*(left)* illustrates the trend of the *First encounter step over episode* during Stage 2. The plot confirms that using trajectories facilitates the robot's finding the person, which it achieves after only approx. 450 steps, while the hGPS and hGPS+traj take more than 700 steps. Finding the robot first

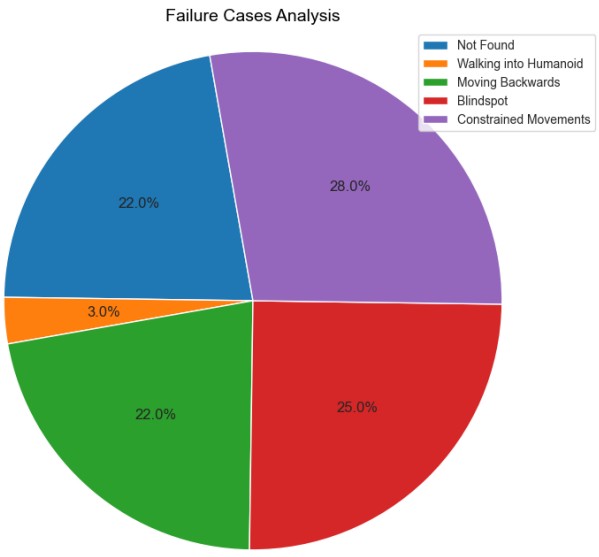

Figure 5: Failure Cases Analysis on 100 episodes

results in higher $S$ and $SPS$ metrics, but it may incur larger chances of collision ($CR$ metric), due to having to then follow it for longer, cf. Sec. 4.1.

Fig. 6*(right)* illustrates the average distance between the agent and the humanoid after the first encounter. While hGPS and hGPS+traj stay at approx. 4 meters, the proposed trajectory-based approach ranges between 1.5 and 1.9, thus well within the 1-2 meter range, which yields the success in the task of following (larger $F$ metric, but encompassing a larger risk of collision– $CR$ metric).

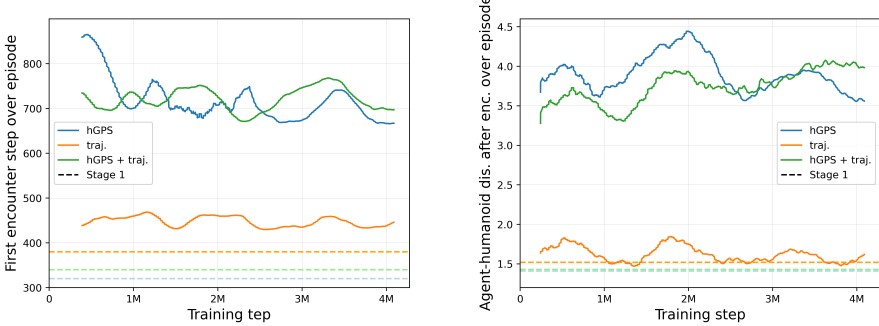

Figure 6: *(left)* Training step (*x*-axis) Vs the number of steps which it takes the robot to find the humanoid (*y*-axis), in the finding task; *(right)* Training step (*x*-axis) Vs the average distance which the robot manages to keep itself at from the humanoid (*y*-axis), in the task of following.

**Design Analysis.** RMA (Kumar et al., 2021) was used to adapt an agent from simulation to real-world deployment. In contrast, SDA encodes human behavior in trajectories, which can be inferred at test time to develop a socially aware navigation policy. While RMA encodes the environment configuration vector $e_t$ with information only at time $t$, we feed an entire trajectory encoding the position in the 20 timesteps up to time $t$. As shown in Table 10, encoding the human position only at time $t$ results in unsatisfactory performance, making the direct application of RMA relatively poor. In fact, it achieves just $3\%$ of Episode Success (ES) against the $43\%$ of SDA.

Given the sequential nature of the states that get encoded, we also evaluated Transformers and MLPs and ultimately selected the MLPs for SDA. In Table 11, we include this preliminary study. Note how the choice of the new sequence modeling mechanism is not trivial and strongly affects performance.

Table 10: Ablation on the length of trajectories to consider during Stage 1

| # States Considered | S ↑ | SPS ↑ | F ↑ | CR ↓ | ES ↑ |
|---|---|---|---|---|---|
| 1 | $0.55^{\pm 0.01}$ | $0.14^{\pm 0.02}$ | $0.03^{\pm 0.01}$ | $0.75^{\pm 0.01}$ | $0.03^{\pm 0.01}$ |
| 5 | $0.62^{\pm 0.01}$ | $0.25^{\pm 0.02}$ | $0.05^{\pm 0.01}$ | $0.64^{\pm 0.01}$ | $0.06^{\pm 0.01}$ |
| 20 (*Proposed*) | $\mathbf{0.92}^{\pm 0.00}$ | $\mathbf{0.46}^{\pm 0.01}$ | $\mathbf{0.44}^{\pm 0.02}$ | $\mathbf{0.61}^{\pm 0.02}$ | $\mathbf{0.50}^{\pm 0.01}$ |
| 50 | $0.70^{\pm 0.02}$ | $0.29^{\pm 0.02}$ | $0.08^{\pm 0.02}$ | $0.78^{\pm 0.02}$ | $0.08^{\pm 0.01}$ |

Table 11: Ablation on the type of encoder to consider during Stage 1.

| Encoder Type | S ↑ | SPS ↑ | F ↑ | CR ↓ | ES ↑ |
|---|---|---|---|---|---|
| MLP (*Proposed*) | $\mathbf{0.92}^{\pm 0.00}$ | $\mathbf{0.46}^{\pm 0.01}$ | $\mathbf{0.44}^{\pm 0.02}$ | $\mathbf{0.61}^{\pm 0.02}$ | $\mathbf{0.50}^{\pm 0.01}$ |
| Transformer | $0.85^{\pm 0.01}$ | $0.15^{\pm 0.02}$ | $0.27^{\pm 0.01}$ | $0.76^{\pm 0.01}$ | $0.12^{\pm 0.01}$ |

## A.9 QUALITATIVE RESULTS

The episode in Fig. 7 (*top*) illustrates the agent's capability to track the human even when it is briefly observed. As the agent searches for the target in the bathroom, the humanoid swiftly passes in front of the door (Step 145) and disappears from the agent's view again (Step 150). Due to learned social dynamics, the agent exploits the humanoid's behavior and begins to follow it. In the episode in Fig. 7 (*bottom*), the agent has already located the human and its objective is to follow it. It is observed that at Step 340, the human decides to move backward, prompting the agent to move backward as well, creating the necessary space for passage by Step 360. Then, it continues its task of following.

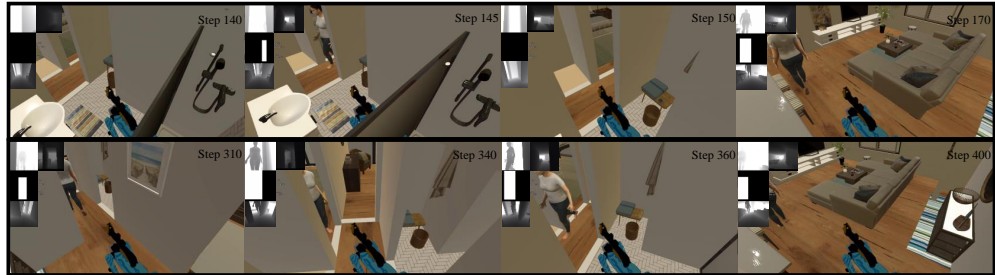

Figure 7: We showcase two different episodes. On the top, the agent successfully follows the human after it swiftly moves in front of the door. On the bottom, an episode where the human decides to move backward and the agent steps back to make way.

