# OpenReview forum: "Following the Human Thread in Social Navigation"
_ICLR.cc/2025/Conference — ICLR 2025 Spotlight_

### Official Review · Reviewer_nz5S · 2024-10-29

**Soundness:** 3
**Presentation:** 2
**Contribution:** 3
**Rating:** 8
**Confidence:** 3

**Summary:**

The paper introduces the Social Dynamics Adaptation model (SDA) to enhance social navigation, where robots must adapt in real-time to human movements. It proposes a two-stage reinforcement learning framework to infer social dynamics from the robot’s state-action history. In the first stage, the model trains using privileged information, like human trajectories, to learn an optimal motion policy. In the second stage, it operates without direct access to human trajectories, instead inferring social dynamics from past actions and statuses.
The approach was tested on Habitat 3.0 and achieved state-of-the-art results in finding and following humans. The primary contributions include the SDA framework and extensive testing under realistic scenarios, which bridge the gap between privileged training data and real-world conditions.

**Strengths:**

1. The SDA model cleverly utilizes privileged information (e.g., human trajectories) during training to learn a social navigation policy and then shifts to infer similar dynamics from historical state-action data during deployment.
2. By modeling human trajectories as latent dynamics, the paper addresses the challenge of partial observability, which is common in robot-centric navigation tasks. The framework effectively models and compensates for missing information, such as losing line of sight of humans, thus proposing to improve the robot's adaptive navigation behavior in dynamic environments.
3. The SDA model is designed to go beyond just avoiding collisions by recognizing and encoding social dynamics, such as keeping an appropriate distance from humans and yielding space when necessary.

**Weaknesses:**

1. The introduction dives too quickly into explaining the simulator, methodology, and related work without clearly stating the problem or why it is hard and important. It lacks motivating examples to contextualize the problem, making it difficult for readers to grasp the significance until section 3.
2. The paper models sensor noise and signal drops to simulate real-world conditions, but this is not sufficient to claim robust generalization to real-world scenarios. Real-world environments involve complex interactions, unpredictability, and physical constraints that are difficult to replicate in simulation, even with added noise.
3. Although the authors test their model extensively in Habitat 3.0, the paper lacks real-world experiments. Simulation platforms, while powerful, cannot fully capture the complexity and unpredictability of real-world environments. Additionally, real-world computational constraints (e.g., processing speed and memory) were not sufficiently addressed.
4. The paper does not sufficiently explore how variability in human behavior, such as different walking speeds, erratic movements, or group interactions, affects the model’s performance.
5. The SDA model focuses primarily on following a single human in simplified indoor environments, without considering more complex scenarios like navigating among multiple humans or handling other social situations.
6. Figure 4 claims that the behavior clusters are well-separated, but they appear mostly overlapping.
7. Figure 2 should explicitly define that stage 1 is offline and stage 2 is online.
8. The paper’s overall presentation could use some work. Many sections of the paper have different writing styles, making the writing really inconsistent, detracting from the message of the paper.

**Questions:**

1. Could you clarify why the problem being solved in this work is hard and important in the introduction? Providing motivating examples early on would help the reader grasp the significance before diving into the methodology, related work or implementation details. Could you revise the introduction to include this context more explicitly?
2. While the paper models sensor noise and signal drops, this may not be enough to claim robust generalization to real-world environments. Could you explain how the model would handle the complexities of real-world social interactions, and consider testing beyond simulated noise scenarios?
3. The paper does not explore how the model adapts to varying human behavior, such as erratic movements. Have you considered testing with more diverse or unpredictable human behaviors? How do you think the model would perform under such circumstances?
4. Given the focus on simulation, could you discuss how the model could be deployed in real-world environments? What are the anticipated challenges of transferring the SDA model to physical robots, and have you considered Sim2Real experiments or alternative testing methods?
5. Could you provide more details on the computational overhead of the two-stage model, particularly during deployment? How do you anticipate the system performing on resource-constrained robots in real-time scenarios, when only stage 2 is run?
6. The current focus is on following a single human in relatively simple environments. Have you considered expanding the model to more complex social settings, such as multi-human interactions?
7. In Figure 4, the text suggests that the behavior clusters are well-separated, but visually, they appear to be overlapping. Could you clarify this discrepancy and perhaps provide a better explanation or adjust the visualization to reflect the claim more accurately?

---

> ### Author Response · Authors · 2024-11-23
>
> - (Q1) **Revised introduction with motivation:**
>   Thanks for raising this point. We agree on the need for a first introductory paragraph to add, which we propose to phrase as follows:
>
>   > Finding a person and following them is relevant to human-robot collaboration. For instance, in search-and-rescue operations, a robot may need to find and reliably follow a human responder through unpredictable and hazardous terrain to assist, carry equipment, or relay critical information. Similarly, in assistive robotics, such as elder care, robots must follow caregivers or patients across different rooms in a home, adapting seamlessly to changes in speed, direction, and environment. The complexity of the task stems from difficult long-term human motion prediction and the possible changes of their intentions, from having to navigate unknown and dynamic environments, and from strict safety requirements.
>
>   These changes are reflected in the main paper. Please note that all updates to the text are highlighted in red for clarity. These highlights will be removed in the final version.
>
> - (Q2) **Sim2Real Transfer:**
>   We thank the reviewer for the feedback and acknowledge the importance of real-world testing. However, this paper's primary focus is demonstrating the validity of our proposed model and its capabilities within simulated environments. This approach aligns with standard practices in Embodied AI research, which commonly rely on simulations to evaluate agent behavior rigorously before real-world deployment [Campari et al. 2020, Cancelli et al. 2023, Chaplot et al. NIPS2020, Chaplot et al. CVPR2020, Ye et al. 2020, Ye et al. 2021, Yokoyama et al. 2022, WACV2024]. Simulations provide controlled and reproducible settings, enabling extensive experimentation and iteration. Furthermore, "Sim2Real" transfer studies [Partsey et al. 2022, ScienceRobotics] are often stand-alone works that bridge the gap between simulated and real-world applications.
>
>   Our work leverages Habitat 3.0, a state-of-the-art simulator, to incorporate sensor noise and signal drops, as described in the main paper (Sec. 5). While not exhaustive, these features capture critical sources of variability encountered in real-world scenarios. Additionally, our adaptation mechanism, which learns latent representations of human motion, has demonstrated resilience to simulated challenges (see the general rebuttal and Table in it), which we have newly performed based on the stimulus of this criticism. These results emphasize the relevance of our simulated evaluations and the robustness of our approach to dynamic changes.
>
>   While focused on simulation, future work could validate our approach in diverse, uncontrolled environments with dynamic human behaviors and complex sensory conditions. Nonetheless, based on numerous tests mimicking increased real-world conditions, the insights presented here remain valuable and transferable, even without real-world experiments.
>
>   - **References:**
>     - [SCIENCEROBOTICS2023] Navigating to objects in the real, Theophile Gervet, Soumith Chintala, Dhruv Batra, Jitendra Malik, and Devendra, Singh Chaplot. *Science Robotics*, 2023.
>     - [WACV2024] Mopa: Modular object navigation with pointgoal agents. Sonia Raychaudhuri, Tommaso Campari, Unnat Jain, Manolis Savva, Angel X Chang. *Proceedings of the IEEE/CVF Winter Conference on Applications of Computer Vision (WACV)*, 2024.

---

> ### Author Response · Authors · 2024-11-23
>
> - (Q3) **Adapting to Unpredictable Human Behaviors**
>   We thank the reviewer for raising this question. Human-robot interaction in real-world scenarios often involves adapting to diverse and unpredictable human behaviors, such as erratic movements. To evaluate the model's adaptability, we conducted experiments simulating varying human motion patterns, including constant, random, and reduced speeds, as summarized in Table 1 of the General Response. These scenarios were designed to mimic real-world conditions, such as following individuals with irregular or slower movements, thereby testing the model's robustness under diverse and dynamic human behaviors.
>
>   - **Performance under constant and random speeds:**
>     The agent showed robust performance across both conditions, maintaining a stable *Finding Success Weighted by Path Steps (SPS)* of 0.45. Similarly, *Finding Success (S)* scores were consistent, with 0.91 achieved for constant speeds and 0.90 for random speeds, demonstrating reliable tracking and navigation efficiency.
>
>   - **Performance under reduced speeds:**
>     When humans moved at slower speeds (constant/2), the agent achieved a *Finding Success (S)* score of 0.87 and significantly reduced its *Collision Rate (CR)* to 0.13, indicating improved safety and adaptability in assisting individuals with reduced mobility.
>
>   - **Impact of unpredictability:**
>     In scenarios with random human speeds, the *Following Rate (F)* decreased from 0.39 to 0.25, reflecting challenges in maintaining optimal following behavior during abrupt and erratic changes in motion. However, this reduction in *Following Rate (F)* is accompanied by an improvement in *Collision Rate (CR)* from 0.57 to 0.48, suggesting that the model adopted a more cautious navigation strategy to minimize risks under unpredictable conditions.
>
>   These findings highlight the model's resilience to variations in human motion patterns, including unpredictability and reduced speeds. While the results demonstrate strong performance in most scenarios, they also indicate areas where the model could be further refined to better handle highly dynamic or erratic behaviors. This adaptability underscores the model's robustness and potential for deployment in real-world, dynamic environments.
>
> - (Q4) **Leveraging Simulation for Robust Policy Training:**
>   Our approach leverages simulation as a safe, cost-effective data source, enabling the training of models using privileged information readily accessible in simulation. During deployment, however, the SDA model relies exclusively on past state-action pairs for adaptation and policy execution, eliminating the need for real-world data collection at runtime. This design enhances practicality and aligns with successful applications in other domains, such as bipedal robot navigation [IROS2022], in-hand object rotations [CoRL2023], and manipulation tasks [CVPR2024]. To address the Sim2Real gap, the main policy could undergo additional fine-tuning in real-world scenarios while keeping the adaptation module frozen, thereby avoiding the need for privileged information during adaptation. Furthermore, discrepancies between simulated and real-world environments—such as noisy or degraded sensor data and unpredictable human movements—can be modeled and incorporated into all training stages of our pipeline through domain randomization. This technique, which introduces stochastic variations in the simulated environment, has proven to be an effective strategy for bridging the Sim2Real gap.
>
>   - **References:**
>     - [IROS2022] Adapting Rapid Motor Adaptation for Bipedal Robots. Ashish Kumar, Zhongyu Li, Jun Zeng, Deepak Pathak, Koushil Sreenath, Jitendra Malik. *IEEE/RSJ International Conference on Intelligent Robots and Systems (IROS)*, 2022.
>     - [CoRL2023] In-Hand Object Rotation via Rapid Motor Adaptation. Haozhi Qi, Ashish Kumar, Roberto Calandra, Yi Ma, Jitendra Malik. *Proceedings of The 6th Conference on Robot Learning (CoRL)*, 2023.
>     - [CVPR2024] Rapid Motor Adaptation for Robotic Manipulator Arms. Yichao Liang, Kevin Ellis, João Henriques. *IEEE/CVF Conference on Computer Vision and Pattern Recognition (CVPR)*, 2024.

---

> ### Author Response · Authors · 2024-11-23
>
> - (Q5) **Computational Efficiency and Deployment Readiness:**
>   While we did not test the model on a physically embodied agent, running it in a simulated environment provides valuable insights into its computational requirements. On a single simulated environment using an NVIDIA RTX 3090 GPU, the model and simulation utilized only 7% of the GPU's computational capacity and 1GB of VRAM. The model consists of approximately 8.8 million parameters and occupies around 35MB of storage, adding only 2MB compared to the baseline architecture. Consequently, its computational demands are directly comparable to the baseline.
>
>   The stage-2 architecture uses the same input as the baseline model and introduces minimal computational overhead, equivalent to the forward pass of the adapter model, which has approximately 99,000 parameters. Given the increasing availability of affordable four-legged robots, we foresee minimal challenges in deploying the model on real-world systems. For instance, the Unitree Go2 EDU robot is available in NANO and NX configurations with NVIDIA Jetson Orin cards. According to the manufacturer's specifications, the NANO cards offer 4GB-40TOPS or 8GB-60TOPS of GPU and AI performance, while the NX cards provide 8GB-70TOPS or 16GB-100TOPS.
>
> Considering that the RTX 3090 GPU, with 35.58 TFLOPS, only required 7% utilization, we estimate that the model should be deployable on most modern and competitive robotic platforms equipped with these embedded GPUs.
>
>
>
> - (Q6) **Adaptability to Multi-Human Interactions and Task Complexity:**
>
> While our current focus is on single-human interactions, our framework is inherently flexible and capable of adapting to more complex social settings involving multiple humans. To provide insights into the performance under such conditions, we evaluated our SDA method and a baseline in environments involving one, two, and three humans, as shown in Table 4. Note that, due to the time allotted to this rebuttal, we did not retrain the policy, but evaluations are based on a policy trained with a single human.
>
> The results highlight the growing difficulty of the task as the number of humans increases. For instance, in single-human scenarios, SDA achieves a success rate (*S*) of **0.91**, which decreases to **0.70** in three-human scenarios. Similarly, the frequency of reaching the target (*F*) drops from **0.39** to **0.12**. These declines are indicative of the increased complexity introduced by needing to navigate around multiple dynamic obstacles while maintaining focus on the target human. The collision rate (*CR*) also rises from **0.57** to **0.78**, further demonstrating the challenges of ensuring safe interactions in crowded spaces.
>
> Despite this increase in task difficulty, our method consistently outperforms the baseline across all metrics and scenarios. For instance, SDA outperforms the baseline in single-human environments by a margin of **0.15** in *S* and **0.10** in *F*. Similarly, in three-human environments, SDA maintains a higher success rate (*S*: **0.70** vs. **0.67**) and frequency of reaching the target (*F*: **0.12** vs. **0.07**). These results demonstrate that while task performance degrades with increased complexity, the SDA framework remains more robust than the baseline.
>
> This performance gap also validates that the observed degradation is a consequence of task complexity rather than a limitation of our method. Moreover, these results are obtained using policies pre-trained exclusively on single-human scenarios, underscoring the adaptability and robustness of our framework even in multi-human environments without additional training.
>
> ---
>
> #### Table 4: Comparison of SDA Performances on Habitat 3.0 with Single and Multiple People
>
> |                          | **S** ↑            | **SPS** ↑         | **F** ↑         | **CR** ↓         | **ES** ↑         |
> |--------------------------|--------------------|--------------------|-----------------|------------------|------------------|
> | **SDA - S2 single-human**   | **0.91 ± 0.01**    | **0.45 ± 0.01**    | **0.39 ± 0.01** | 0.57 ± 0.02      | **0.43 ± 0.02**  |
> | Baseline single-human      | 0.76 ± 0.02        | 0.34 ± 0.01        | 0.29 ± 0.01     | **0.48 ± 0.03**  | 0.40 ± 0.02      |
> | **SDA - S2 two-humans**     | 0.80 ± 0.01        | 0.41 ± 0.02        | 0.18 ± 0.01     | 0.68 ± 0.01      | 0.20 ± 0.01      |
> | Baseline two-humans        | 0.76 ± 0.01        | 0.36 ± 0.02        | 0.09 ± 0.01     | 0.80 ± 0.01      | 0.11 ± 0.01      |
> | **SDA - S2 three-humans**   | 0.70 ± 0.01        | 0.38 ± 0.02        | 0.12 ± 0.01     | 0.78 ± 0.01      | 0.15 ± 0.01      |
> | Baseline three-humans      | 0.67 ± 0.01        | 0.36 ± 0.02        | 0.07 ± 0.01     | 0.83 ± 0.01      | 0.09 ± 0.01      |

---

> ### Author Response · Authors · 2024-11-23
>
> - (Q7) **Clarifying Figure 4 and behavior clusters:**
>
>     We appreciate the reviewer’s insightful observation regarding the overlap between clusters in Figure 4. The updated visualization in the rebuttal's PDF provides a clearer depiction of how behavioral stages are distributed in the latent space, incorporating both time progression and cluster-specific scatter plots. Please note that all updates to the text are highlighted in red for clarity. These highlights will be removed in the final version.
>
>     From the newly-colored scatter plot, it is now clearer that the only overlapping clusters are those associated with behaviors that share inherent similarities. For instance, behaviors such as Seek and Follow involve actively observing and responding to the human’s position, naturally leading to shared characteristics in the latent space. Likewise, transitions between stages like Lost and Seek are continuous rather than sharply defined, resulting in overlapping regions in the visualization. This observation aligns with the nature of human-robot interactions, where behaviors often transition fluidly rather than being strictly discrete.

---

> > ### Comment · Reviewer_nz5S · 2024-11-24
> > **Response to Authors' Rebuttal**
> >
> > The authors address all my concerns about the paper through their detailed and thorough rebuttal. I think these changes will make the paper a lot better. I specifically find it commendable that they conducted new experiments and found greater support for the efficacy of their approach. Therefore I am increasing my rating to Accept.

---

### Official Review · Reviewer_Naey · 2024-11-04

**Soundness:** 3
**Presentation:** 4
**Contribution:** 3
**Rating:** 8
**Confidence:** 3

**Summary:**

This paper introduces the Social Dynamics Adaptation (SDA) model, a two-stage reinforcement learning approach for enhancing robot social navigation. SDA aims to improve a robot’s ability to navigate alongside humans in dynamic environments, where human trajectories serve as social cues that guide robot movement. In the first stage, the model learns a policy using privileged information about human trajectories, encoding social dynamics into a latent representation. In the second stage, this policy operates without access to these trajectories, relying on inferred social dynamics from the robot’s own state and action history.
Experiments conducted on the Habitat 3.0 platform demonstrate that SDA significantly outperforms state-of-the-art baselines in tracking, following, and maintaining appropriate distances from humans. The study includes analyses of SDA’s effectiveness under noisy conditions, different update rates, and with realistic humanoid behaviors, showcasing its potential for adapting to real-world conditions.

**Strengths:**

Firstly, the SDA model’s two-stage training framework effectively combines privileged information during training with adaptable inferences during deployment, allowing the model to handle real-world scenarios without needing precise human trajectory data. It outperforms several baselines, showing improved metrics for finding, following, and episode success, which is critical for effective social navigation. Testing on Habitat 3.0, with realistic human motion and environmental noise, strengthens the claim that SDA is robust and adaptable to real-world applications. Comprehensive evaluations, including noise analysis, reduced update frequencies, and alternative metrics, provide a deep understanding of the model’s strengths and weaknesses under various conditions.

**Weaknesses:**

Despite testing under realistic noise and movement in simulation, SDA’s real-world performance remains untested, raising questions about its transferability from simulated to physical environments. Besides, the model assumes a certain pattern of human movement and interaction, which may limit its effectiveness in scenarios where human behaviors deviate significantly from training data.

**Questions:**

Are there any plans to test SDA in real-world environments to evaluate how well it transfers from simulation, particularly in handling variable lighting, unexpected obstacles, and diverse human behaviors?
How does SDA perform when humans change speed or direction quickly? Does the reliance on past state-action history allow the model to adapt in real-time, or are there latency issues?

---

> ### Author Response · Authors · 2024-11-23
>
> - **Reducing the Sim-to-Real Gap:**
>
> We acknowledge the reviewer’s concern regarding the transferability of SDA from simulation to real-world environments. Human-robot cooperation will inevitably require real-world testing as advancements in modeling and safety progress. However, in this work, the progress made by SDA over the state-of-the-art (SOTA) will likely remain valid in real-world applications. This is supported by the diverse experiments that mimic increasing aspects of real-world complexities, including sensor noise, erratic human motion, communication, and processing latency.
>
> Specifically, our experiments in the Habitat 3.0 simulator were designed to replicate key real-world challenges, such as sensor imperfections, signal drops, and variability in human motion patterns. For example, SDA's performance remained robust despite these introduced complexities, demonstrating resilience to diverse and unpredictable behaviors. This indicates that the modeling updates provided by SDA—particularly its emphasis on learning latent representations of human trajectories and its ability to adapt to dynamic obstacles—represent a significant advancement over the baseline and current SOTA, with strong potential for real-world transferability.
>
> - **Simulating Changing Human Motion Patterns:**
>
> We thank the reviewer for the detailed question. Human-robot cooperation requires adapting to diverse and unpredictable behaviors in real-world settings. To address this, we designed experiments to evaluate SDA's performance under scenarios with varying human motion patterns, including constant, random, and reduced speeds, as shown in Table 1 (General Response). These experiments aim to simulate real-world conditions, such as assisting individuals with reduced mobility or adapting to unpredictable human behaviors. Note that, due to the time allotted to this rebuttal, we did not retrain the policy, but evaluations are based on a policy trained with constant speed.
>
> - **Performance under constant and random speeds:** The agent demonstrated consistent navigation efficiency, with *Finding Success Weighted by Path Steps (SPS)* remaining at 0.45 across both conditions. The *Finding Success (S)* scores were similarly robust, achieving 0.91 for constant speeds and 0.90 for random speeds.
>
> - **Performance under reduced speeds:** When humans moved at slower speeds (constant/2), the agent achieved a *Finding Success (S)* score of 0.87 and a reduced *Collision Rate (CR)* of 0.13, highlighting improved safety under such conditions.
>
> - **Impact of unpredictability:** In the case of random human speeds, the *Following Rate (F)* decreased from 0.39 to 0.25, indicating challenges in adapting to abrupt changes in human motion. However, this decline in *Following Rate (F)* corresponded with an improvement in *Collision Rate (CR)* from 0.57 to 0.48, suggesting that the model adopted a more cautious navigation strategy in response to unpredictable behaviors.
>
> These results underscore SDA's resilience to variations in human motion while revealing areas for improvement, particularly in handling highly dynamic and erratic behaviors.
>
> - **SDA is Robust to Latency Issues:**
>
> SDA relies on its adaptation module to learn latent representations of human motion, allowing it to respond effectively to abrupt changes in speed or direction. To evaluate its real-time adaptability, we tested the model’s robustness to varying sensor update rates, as detailed in Table 4 of the main paper. These tests simulated scenarios with reduced update frequencies, mimicking real-world latency.
>
> The results indicate that SDA maintains comparable performance even under delayed updates, suggesting that its reliance on past state-action history does not introduce significant latency issues under realistic conditions. This robustness supports its practicality in dynamic, real-world environments.

---

> > ### Comment · Reviewer_Naey · 2024-11-27
> >
> > Thank you for the detailed explanation. After reconsidering the paper including your responses, my overall assessment remains unchanged.
> > For the first question, while such simulated evaluations are an important step, real-world testing is essential to truly validate transferability. Hardware-related constraints often introduce challenges that are difficult to simulate entirely.
> > For the second question, the results provided for varying human motion patterns are valuable and insightful. Including these findings as an ablation study would enhance the paper's quality and provide a more comprehensive understanding of the model's performance under diverse conditions.
> > For the third question, I appreciate the clarity of your response and the additional details provided.
> > Overall, it is a good paper with meaningful contributions, and I believe it has great potential for future impact.

---

### Official Review · Reviewer_Cwf1 · 2024-11-06

**Soundness:** 3
**Presentation:** 3
**Contribution:** 2
**Rating:** 6
**Confidence:** 3

**Summary:**

This paper proposes a new method enabling a robot to search for a human in a simulated indoor environment and then follow them by maintaining a distance of 1 to 2 meters for at least k steps.

The key challenge is that the robot can only see the human via its first-person egocentric camera, resulting in partial knowledge about the environment, as well as the position and velocity of the human.

To address this problem, the authors propose to rely on privileged information, e.g., human GPS data and human trajectories, that are only available during training, not deployment. Then, the authors propose to train the robot policy in 2 phases. In phase 1, the robot does RL (DD-PPO) to jointly learn an encoder that maps privileged information to the latent state and a policy that depends on the robot camera image and the latent state. In phase 2, the robot will run the learned policy while doing supervised learning to learn a decoder to map the robot's camera image to the latent state (the data for the latent state comes from the ground truth privileged information, and the learned encoder from phase 1). Then, during online deployment, the robot will first convert the camera image to the latent state using the decoder from phase 2, and then run the robot policy learned from phase 1.

The result focuses on the Habitat 3.0 simulator. The proposed method is shown to outperform 3 baseline methods. An ablation study was conducted to evaluate the usefulness of each privileged information. Latent analysis shows that the latent state refers to the robot's subtask in find, seek, lost, and follow. The authors also add more realistic components into the simulator to simulate real-world scenarios, such as making the simulated human aware of the robot's presence with reciprocal collision avoidance (ORCA), lowered planning frequency, and increased noise. In such scenarios, the proposed method still outperforms the baseline methods.

**Strengths:**

- The problem is well-motivated.
- The writing is clear.
- The method is sound.

**Weaknesses:**

- As far as I can tell, the training and testing assume the same simulated person. In the real world, this means that to follow one person, the robot has to interact with the user to collect offline data for Phase 1 and online data for Phase 2. This can be very cumbersome. It would be great if the authors could discuss some strategies to scale this method up for different people, such as using meta-learning approaches or using more diverse training data.

- The result shown in Tab.1 shows only marginal improvement over the baseline "Baseline+Proximity Cancelli et al. (2023)". Could the authors provide an discussion about the practical implications of these improvements? Another way to demonstrate its significance is to use statistical test.

**Questions:**

- Regarding technical novelty, the authors discuss the comparison between the proposed work and the previous work (Cancelli et al., 2023) in Sec.2. It seems that the techniques in both works are similar, but this paper proposes to use a similar structure for the following-human problem, rather than just avoiding collisions with humans as in Cancelli et al., 2023. It would be great if the author could clarify about the difference between these work, or about the extra challenges introduced by following a person rather than just collision avoidance (e.g., perhaps the challenge is more than just changing the reward function of the RL method?).

---

> ### Author Response · Authors · 2024-11-23
>
> - (W1) **Enhancing Realism in Simulated Human Motion:**
> We appreciate the reviewer’s interesting statement. Currently, Habitat provides encoded elements, such as target destinations, which are sampled to generate scenarios. However, we agree that leveraging advanced models to simulate specific human motion patterns learned from data is an exciting and valuable direction. There are now several state-of-the-art methods for motion synthesis, including simulating personalized motion styles or interactions within 3D scenes. For example:
>     - **Sampieri et al. [ECCV24Sampieri]** propose a length-aware motion synthesis method using latent diffusion.
>     - **Dai et al. [ECCV24Dai]** present MotionLCM, enabling real-time controllable motion generation.
>     - **Jiang et al. [CVPR24Jiang]** and **Wang et al. [CVPR24Wang]** demonstrate scalable approaches for modeling dynamic human-scene interactions and affordance-driven motion generation.
>
>   We believe incorporating such methods into simulation frameworks would enhance the realism and diversity of human motion, further enriching research in this area and aligning with our vision for future improvements.
>
>   We also appreciate the suggestion of using meta-learning to unify Phase 1 and Phase 2 and reduce overall training time. Phase 1 requires approximately four training days, while Phase 2 takes just two hours, making the latter a minimal effort. Meta-learning could be a valuable approach, particularly for learning specific individuals and their unique motion patterns in randomly generated scenes. For example, one could imagine training a general meta-human motion model that can be fine-tuned to adapt to specific scenarios. These ideas are highly promising and align with our vision for future work, where we aim to enhance the scalability and adaptability of our method.
>
> #### References:
> - [ECCV24Sampieri] Sampieri, A., Palma, A., Spinelli, I., and Galasso, F. (2024). *Length-Aware Motion Synthesis via Latent Diffusion*. 2024 European Conference on Computer Vision (ECCV).
> - [ECCV24Dai] Dai, Wen-Dao et al. *MotionLCM: Real-time Controllable Motion Generation via Latent Consistency Model.* 2024 European Conference on Computer Vision (ECCV).
> - [CVPR24Jiang] Jiang, Nan et al. *Scaling Up Dynamic Human-Scene Interaction Modeling.* 2024 IEEE/CVF Conference on Computer Vision and Pattern Recognition (CVPR).
> - [CVPR24Wang] Wang, Zan et al. *Move as you Say, Interact as you can: Language-Guided Human Motion Generation with Scene Affordance.* 2024 IEEE/CVF Conference on Computer Vision and Pattern Recognition (CVPR).

---

> ### Author Response · Authors · 2024-11-23
>
> - (W2) **Statistical Significance and Robustness of SDA:**
>   The proposed SDA method extends the baseline [34], rather than Cancelli et al. [8], and does so consistently across all metrics. Specifically, as shown in Table 1 of the paper, SDA improves on the primary tasks of:
>     - Finding the human (*Finding Success*, S),
>     - Reaching the human using an optimal path (*SPS*), and
>     - Following the human (*F*).
>
>   These improvements are by 15, 11, and 10 percentage points (p.p.), respectively (from 76% to 91% on S, from 0.34 to 0.45 on SPS, and from 0.29 to 0.39 on F). While the overall improvement in *Episode Success* (ES) is more modest (from 0.41 to 0.43), this is primarily due to the increased *Collision Rate* (CR). As discussed in line 347 of the main paper, SDA finds the human earlier and follows them for much longer—on average, 390 steps instead of 218. This extended following time poses a more significant challenge for SDA to avoid collisions, leading to a higher CR and moderating its margin of improvement in ES.
>
>   Cancelli et al. [8] extend Habitat [25] with research directions that are, in some sense, orthogonal to our work. While the performance of Cancelli et al. [8] appears similar to SDA in some metrics, SDA demonstrates a significantly longer follow time, highlighting its robustness in maintaining the task objective over extended durations. Additionally, Table 1 includes a third comparison technique, Baseline+Proximity [8], which is state of the art for social navigation in simulators [25, 36]. We adapted [8] to Habitat 3.0 for comparison, and while it outperforms the baseline, SDA surpasses [8] across all reported metrics. Certain complementary aspects of [8] could be integrated into SDA to enhance performance further; however, such extensions fall outside the scope of this work.
>
>   To evaluate the statistical significance of the reported improvements, we performed independent two-sample *t*-tests assuming unequal variances (Welch's *t*-test). This statistical analysis confirms that the improvements of SDA over both the baseline and Cancelli et al. [8] are significant, further supporting our claims regarding the effectiveness of the proposed method.
>
> ---
>
> #### Table 2: Updated t-statistics and p-values for the comparison of methods.
>
> | **Metric** | **Baseline + Proximity** | **SDA - S2** | **t-statistic** | **p-value** | **Significance** |
> |------------|---------------------------|--------------|-----------------|-------------|------------------|
> | S          | 0.85 ± 0.02               | 0.91 ± 0.01  | -8.49           | 0.00000103  | Significant       |
> | SPS        | 0.41 ± 0.02               | 0.45 ± 0.01  | -5.66           | 0.0000732   | Significant       |
> | F          | 0.37 ± 0.01               | 0.39 ± 0.01  | -4.47           | 0.000295    | Significant       |
> | CR         | 0.58 ± 0.02               | 0.54 ± 0.02  | 4.47            | 0.000295    | Significant       |
> | ES         | 0.42 ± 0.01               | 0.45 ± 0.02  | -4.24           | 0.000924    | Significant       |
>
> ---
>
> The statistical significance results for each pair of values are presented in Table 2. Below is the interpretation for each comparison:
>
> - The difference between 0.85 and 0.91 is statistically significant (*P = 0.00000103*).
> - The difference between 0.41 and 0.45 is statistically significant (*P = 0.0000732*).
> - The difference between 0.37 and 0.39 is statistically significant (*P = 0.000295*).
> - The difference between 0.58 and 0.54 is statistically significant (*P = 0.000295*).
> - The difference between 0.42 and 0.45 is statistically significant (*P = 0.000924*).

---

> ### Author Response · Authors · 2024-11-23
>
> - (Q1) **Key Differences Between SDA and Cancelli et al. (2023)**
>
> The **key difference** between SDA and Cancelli et al. (2023) lies in the **methodological design** and how tasks are addressed. While Cancelli et al. can be adapted to Find and Follow tasks, their approach is fundamentally **reactive**, relying on **Proximity Tasks** (Risk Estimation and Proximity Compass) to estimate local collision risks and directional guidance. These tasks serve as auxiliary signals during training, helping the agent make short-term decisions based on immediate sensory input.
>
> #### Table 3: Comparative Overview of Methodological Differences
>
> | **Feature**              | **Cancelli et al. (2023)**                     | **SDA**                                           |
> |--------------------------|-----------------------------------------------|--------------------------------------------------|
> | **Core Approach**        | Proximity-based auxiliary tasks (Risk, Compass) | Two-stage predictive framework (latent dynamics model) |
> | **Decision-Making**      | Reactive, short-term focus                     | Predictive, anticipates human behavior           |
> | **Training Stages**      | Single-stage end-to-end with auxiliary tasks   | Two-stage: privileged learning and adaptation    |
> | **Inference Input**      | Local sensory data (proximity, direction)      | History of states and actions (no privileged data) |
> | **Behavioral Modeling**  | Local collision avoidance                      | Full social dynamics (intentions and trajectories) |
>
> ---
>
>
> By contrast, SDA introduces a **predictive, two-stage approach** that shifts the focus from reactive to **anticipatory behavior**:
> 1. **Stage 1:** SDA uses privileged trajectory information to encode latent social dynamics into a compact representation, which informs the motion policy.
> 2. **Stage 2:** The agent adapts to infer these dynamics solely from past actions and observations, enabling proactive decision-making even without direct trajectory data.
>
> This design allows SDA to better generalize to dynamic scenarios where human behavior is less predictable.
>
> Additionally:
> - **Cancelli et al.** rely heavily on proximity-based signals, which are effective for **collision avoidance** but limited in capturing broader social dynamics.
> - **SDA**, by explicitly modeling **social dynamics**, enables the agent to anticipate and align with **human intentions**, balancing proximity and continuity for tasks like human following.
>
> This predictive capability is essential for maintaining robust social navigation. These distinctions will be better clarified in the final version of the paper.

---

### Author Response · Authors · 2024-11-23
**General Response and Experiment on Changing Human Motion Patterns**

We are deeply thankful for the appreciation of several aspects of this work. We also appreciate the reviewers' efforts to formulate thoughtful and constructive criticism. Having addressed it, we feel the paper is now a more solid contribution.

We report in this general response novel experiments motivated by the reviewers' criticism which complement the arguments and experiments in the original submission.


### Simulating Changing Human Motion Patterns

Following the suggestions of reviewers Naey and nz5S, we simulate varying motion patterns of humans, accounting for cases where they could abruptly change their minds and walk at reduced speed due to impediments or reduced mobility. These experiments complement the ablation studies on sensor noise and processing latency (see Section 5 in the submitted manuscript). Note that, due to the time allotted to this rebuttal, we did not retrain the policy, but evaluations are based on a policy trained with constant speed.

---

#### Table 1: Comparison of SDA Stage 2 performances using constant human speed (h.speed) or random human speed in the simulator.

| Model            | h.speed    | S ↑           | SPS ↑         | F ↑           | CR ↓         | ES ↑          |
|-------------------|------------|---------------|---------------|---------------|---------------|---------------|
| **SDA - S2**    | constant   | **0.91 ± 0.01** | **0.45 ± 0.01** | **0.39 ± 0.01** | 0.57 ± 0.02  | 0.43 ± 0.02   |
| **SDA - S2**    | constant/2 | 0.87 ± 0.01    | 0.57 ± 0.02    | 0.28 ± 0.01    | **0.13 ± 0.01** | **0.70 ± 0.01** |
| **SDA - S2**    | random     | 0.90 ± 0.01    | **0.45 ± 0.02** | 0.25 ± 0.01    | 0.48 ± 0.01  | 0.40 ± 0.01   |

---

Let us consider the novel evaluations in Table 1:

- **Performance under constant and random speeds:** The agent maintained consistent navigation efficiency, with *Finding Success Weighted by Path Steps (SPS)* remaining at 0.45. The *Finding Success (S)* scores were also comparable: 0.91 for constant speeds and 0.90 for random speeds.
- **Performance under reduced speeds:** For slower human movement (constant/2), the agent achieved a *Finding Success (S)* score of 0.87 and a reduced *Collision Rate (CR)* of 0.13, indicating improved safety.
- **Impact of unpredictability:** Under random speeds, the *Following Rate (F)* decreased from 0.39 to 0.25. This suggests challenges in adapting to abrupt changes in human motion. Interestingly, the decrease in *Following Rate (F)* corresponded with an improvement in *Collision Rate (CR)* from 0.57 to 0.48, reflecting a cautious strategy in response to unpredictable behaviors.

These results highlight the model’s resilience to human speed variations while revealing areas for improvement, particularly in handling dynamic and erratic behaviors.

---

### Note · Authors · 2025-01-30

**Comment:**

We would like to thank the reviewers for their thoughtful feedback and valuable suggestions. After careful consideration, we have decided to withdraw our submission in order to take the necessary time to address the points raised and further improve the manuscript.

**Withdrawal Confirmation:**

I have read and agree with the venue's withdrawal policy on behalf of myself and my co-authors.

---

> ### Note · Program_Chairs · 2025-02-05
>
> **Comment:**
>
> Requested by authors.
>
> **Revert Withdrawal Confirmation:**
>
> We approve the reversion of withdrawn submission.

---

### Decision · Program_Chairs · 2025-01-22

Accept (Spotlight)